# Reactive P and S co-doped porous hollow nanotube arrays for high performance chloride ion storage

Siyang Xing[1,2], Ningning Liu[1], Qiang Li[1], Mingxing Liang[1,3,4], Xinru Liu[1,5], Haijiao Xie [6], Fei Yu[7] & Jie Ma [1,8] ✉

Developing stable, high-performance chloride-ion storage electrodes is essential for energy storage and water purification application. Herein, a P, S co-doped porous hollow nanotube array, with a free ion diffusion pathway and highly active adsorption sites, on carbon felt electrodes (CoNiPS@CF) is reported. Due to the porous hollow nanotube structure and synergistic effect of P, S co-doped, the CoNiPS@CF based capacitive deionization (CDI) system exhibits high desalination capacity (76.1 $mg_{Cl^-}$ $g^{-1}$), fast desalination rate (6.33 $mg_{Cl^-}$ $g^{-1}$ $min^{-1}$) and good cycling stability (capacity retention rate of > 90%), which compares favorably to the state-of-the-art electrodes. The porous hollow nanotube structure enables fast ion diffusion kinetics due to the swift ion transport inside the electrode and the presence of a large number of reactive sites. The introduction of S element also reduces the passivation layer on the surface of CoNiP and lowers the adsorption energy for Cl⁻ capture, thereby improving the electrode conductivity and surface electrochemical activity, and further accelerating the adsorption kinetics. Our results offer a powerful strategy to improve the reactivity and stability of transition metal phosphides for chloride capture, and to improve the efficiency of electrochemical dechlorination technologies.

Sustainable clean energy, water scarcity and industrial wastewater treatment demands have prompted the research of chloride ion storage, which has also been widely studied in the fields of rechargeable chloride ion batteries[1,2] and desalination[3,4]. Thus it is imperative to find effective and reliable technologies for storage of excessive chloride from water and wastewater[3]. Although the traditional techniques can separate chloride ions from the reclaimed water, such as precipitation[5], electrolysis[6], ion exchange[7], and evaporative concentration[8], they have certain drawbacks such as high energy consumption, high cost, and complex maintenance processes[9].

Chloride-ion capture for CDI has gained significant attraction in recent years as a promising water-treatment technology to tackle water shortage and chloride pollution problems[10]. Apart from its low cost and easy operation, CDI has great potential as a platform for a combination of freshwater access and energy recovery[11–13].

Electrodes are the core component of electrochemical desalination and ion separation. Up to now, only a few Cl⁻ capture materials

[1]Research Center for Environmental Functional Materials, State Key Laboratory of Pollution Control and Resource Reuse, College of Environmental Science and Engineering, Tongji University, Shanghai 200092, PR China. [2]Department of Energy, Environmental & Chemical Engineering, Washington University in St. Louis, St. Louis, MO 63130, USA. [3]College of Chemistry and Environmental Engineering, Shenzhen University, Shenzhen 518060, PR China. [4]College of Physics and Optoelectronic Engineering, Shenzhen University, Shenzhen 518060, PR China. [5]School of Architecture, Civil and Environmental Engineering, EPFL, Lausanne Vaud1015, Switzerland. [6]Hangzhou Yanqu Information Technology Co., Ltd., Y2, 2nd Floor, Building 2, Xixi Legu Creative Pioneering Park, No. 712 Wen'er West Road, Xihu District, Hangzhou 310003, PR China. [7]College of Oceanography and Ecological Science, Shanghai Ocean University, No 999, Huchenghuan Road, Shanghai 201306, PR China. [8]School of Civil Engineering, Kashi University, Kashi 844000, PR China. ✉e-mail: jma@tongji.edu.cn

have been developed, including Ag/AgCl[14,15], Bi/BiOCl[16,17], Layered Double Hydroxides (LDHs)[18–20], conductive polymers[21–23], etc. However, the application of Ag/AgCl is limited by its high cost and strong volume changes in reversible reactions; Bi/BiOCl is restricted by its sluggish kinetics, high energy consumption and electron consumption in disequilibrium with cathodic adsorption[4]; LDHs has relatively insufficient conductivity and low cycling stability[18]; conductive polymer electrodes have a low desalination capacity, currently the highest capacity is around 20 mg g$^{-1}$ [23]. As a consequence, it is very urgent to exploit high-performance electrodes for advancing Cl$^-$ storage.

Transition metal phosphides (TMPs), formed through combining phosphorus with transition metal elements, are considered among the most advantageous electrode materials for CDI. This is attributed to their high electrochemical activity, excellent electrical conductivity, and metalloid properties[24–26]. In contrast to TMOs or LDHs, TMPs demonstrate higher specific capacity and superior electrical conductivity. The distinct nature of phosphides come from their lower electronegativity (Pauling electronegativity: 3.44 for O and 2.19 for P) and larger atomic radius (0.074 nm for O and 0.109 nm for P)[27]. This results in desirable physicochemical and electrochemical characteristics[28]. Despite the excellent electrochemical attributes of pristine TMPs, they are still constrained by certain limitations in practical application in CDI system: (1) the electrode capacity is hindered by its sluggish reaction kinetic, (2) the electrode stability is poor due to the volume expansion during cycling, (3) the electrode may reconstitute to form unstable phases in neutral and/or alkaline environments, such as transition metal hydroxide[29]. Therefore, slow reaction kinetics and instability are some serious issues that need to be addressed[30].

To address these issues, various strategies have been devised to improve the performance of TMPs electrodes. Chemical Vapor Deposition (CVD) method has been utilized as a general strategy for P-doping process. Annealing temperature has a crucial effect on the growth and crystalline of particles. The low annealing temperature may decrease the crystalline, while a high temperature could result in the aggregation of nanoparticles, reducing the surface areas[31]. The Ni-Fe-P-350 nanosheets prepared by Liu et al. at the optimum temperature possess a superior specific surface area of 75 m$^2$ g$^{-1}$ and deliver a highest capacity of 1358 C g$^{-1}$ at 5 mA cm$^{-2}$ [32]. In addition, the combination of TMPs with carbon improves its electrical conductivity and the flexible structure of carbon reduces the chalking of the electrodes caused by the expansion of the TMPs volume[33]. Besides, doping or vacancies are effective ways to adjust electronic structure, increase surface reactivity, and generate more surface active sites, thereby enhancing capacity[34]. The introduction of oxygen vacancies into cobalt molybdate (CoMoO$_4$) crystal lattices by using different P source dosages shows higher electrical conductivity ($3.9 \times 10^{-2}$ S m$^{-1}$) than pristine CoMoO$_4$ ($5.7 \times 10^{-3}$ S m$^{-1}$), and higher surface redox activity, increasing redox reaction species and facilitating electron transport[35]. However, a single phosphorus vacancy defect can also cause crystal geometry deformation and symmetry breaking, triggering structural defects and generating hole traps, causing the crystal structure to be destroyed during the reaction, leading to a decrease in stability[36,37]. The sluggish reaction kinetics and volumetric expansion during charging/discharging progress remain as a bottleneck hindering the capacity and cycling stability of TMPs capacitors. In addition, in the removal of chloride ions, exploring suitable ion diffusion conditions is an important requirement for improving diffusion kinetics.

Herein, CoNiPS@CF with high-speed diffusion channels and highly active sites is obtained by gradually P-doping and S-dpoing CoNiOH nanoarrays grown in situ on the surface of pretreated carbon felt (pCF) and is used as the anode in hybrid capacitive deionization (HCDI) for Cl$^-$ removal, which has been largely overlooked in relevant research. The synergistic effect of bimetallic Co-Ni phosphides can ameliorate the CDI performance, with a widened potential window, improved electric conductivity, and activated reaction sites[38]. P doped

CoNiP@CF electrode can significantly improve the conductivity of the precursor, achieving high the electron transport ability and ion adsorption kinetics. To improve the poor stability of TMPs and increase the adsorption capacity, CoNiP@CF was further doped with sulfur to form a porous hollow nanoarray CoNiPS@CF electrode. By doping S into TMPs, high-valent metal centers can be formed, facilitating interfacial charge transfer and enhancing the redox activity capability of active species[39], which is benefited from electron density distortion caused by high electronegativity of S atom. Furthermore, porous hollow nanotube structure possesses a large number of surface-active sites and high-speed ion transport channels which facilitate high-rate and high-capacity Cl$^-$ adsorption; the hollow structure reduces expansion of the bulk phase caused by Cl$^-$ adsorption, therefore improving cycling stability. CoNiPS@CF electrode combining the synergistic effect of hollow structure and P, S co-doped exhibited superior CDI performance with a high desalination capacity of 76.1 mg$_{Cl^-}$ g$^{-1}$, a fast desalination rate of 6.33 mg$_{Cl^-}$ g$^{-1}$ min$^{-1}$, and excellent cycling stability with a capacity retention greater than 90%, which surpassed the other state-of-the-art CDI electrodes containing carbonaceous and faradaic materials under similar experimental conditions. This work opens a avenue for increasing the rate of reaction and opening of active sites at Faraday electrodes, as well as for the improvement of the reactivity and stability of transition metal phosphides, which can be used in advanced dechlorination techniques.

## Results

### Synthesis and characterization of CoNiPS@CF

The growth of CoNiP nanoneedles and CoNiS nanotubes on pCF was shown in Fig. 1a (more details are in Methods). Typically, CoNiOH nanoneedles were first grown in situ on pCF by hydrothermal method and then thermally annealed at 300 °C for 2 h in the presence of Ar atmosphere and phosphorus source to form CoNiP. To introduce sulfur atoms, the prepared CoNiP nanoneedles were further thermally annealed at 400 °C for 1 h in the presence of Ar atmosphere and sulfur source to form CoNiPS. The SEM images of CoNiOH@CF, CoNiP@CF and CoNiPS@CF were shown in Fig. 1b–d, respectively. The inserted low magnification reduced images showed the CoNiP nanoneedle and CoNiPS nanotube arrays growing uniformly over the entire surface of the pCF. The high-magnification SEM images showed that the needle morphology was retained after the P-doping and S-doping processes. Supplementary Fig. 1 and Fig. 1e showed the elemental distribution of CoNiP@CF and CoNiPS@CF. The Co, Ni and P elements are evenly distributed on the surface of the CoNiP@CF electrode, indicating that P was successfully and uniformly doped into the electrode during the P-doping process. In the further S doping process, both S and P elements were uniformly doped on the fiber surface, indicating that S and P were co-doped on the electrode. In addition, except for the original carbon felt (148.2°), the water contact angle of other samples is 0°, showing excellent hydrophilicity (Supplementary Fig. 2).

The structural details of CoNiP@CF and CoNiPS@CF were further examined through transmission electron microscopy (TEM, in Fig. 1f, h). The EDS line scan of Fig. 1g showed that the central region element content is high, while the edges were low, indicating that the P-doping process basically did not change the nanoneedle morphology. The CoNiPS material on the surface of pCF becomes hollow, the elements in the middle and at the edges of the nanotubes was basically the same (Fig. 1i). Besides, there were nanopores on the surface of the CoNiPS nanotubes (red circles in Fig. 1h). The Kirkendall effect could explain how nanotube structures form[40,41]. S$^{2-}$ first reacted with CoNiP to form a thin layer of CoNiPS, which could reduce the reaction rate between the external S$^{2-}$ and the internal CoNiP as a barrier. Due to the non-equilibrium diffusion process between inward S$^{2-}$ and outward CoNiP, voids were created in the center of the nanoneedles. During the progression of the reaction, the CoNiPS shell thickens and the CoNiP nuclei gradually decrease, eventually forming CoNiPS nanotubes;

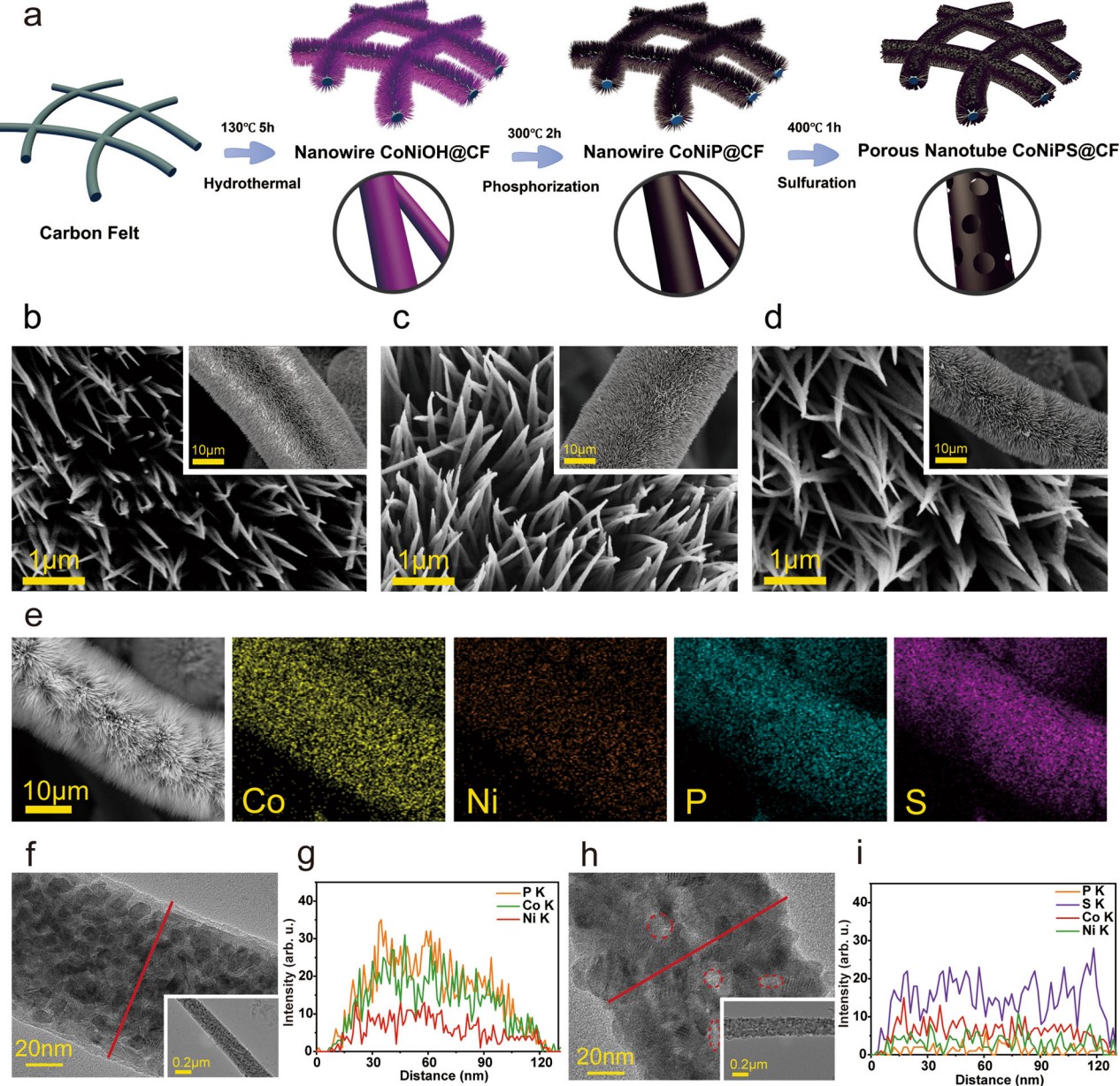

**Fig. 1 | Synthesis and hollow structural characterizations of CoNiPS@CF and CoNiP@CF.** Schematic diagram of the preparation process of (**a**) P, S double-doped CoNiPS@CF hollow nanoarrays, high magnification and low magnification (the inset) SEM images of (**b**) CoNiOH@CF, (**c**) CoNiP@CF and (**d**) CoNiPS@CF electrodes, and **e** CoNiPS@CF corresponding elemental distribution maps. **f**, **h** High magnification and low magnification (the inset) TEM images and (**g**, **i**) compositional line profiles by EDS line scanning of CoNiP@CF and CoNiPS@CF. The red circles represent nanopores on the structure.

while the CoNiP nuclei present at the surface likewise become nano-centers, forming new voids on the surface of the nanotubes with external $S^{2-}$ non-equilibrium diffusion, thus creating surface pores. The diameter of the CoNiP@CF nanoneedles and CoNiPS@CF nanotubes is about $86.73 \pm 12.94$ nm and $106.64 \pm 16.6$ nm, which also proves the formation of CoNiPS porous nanotubes (Supplementary Fig. 3).

Figure 2a showed the XRD patterns of the CoNiP@CF and CoNiPS@CF samples. The peak of CoNiP@CF was consistent with the standard card JCPDS 29-0497, indicating the successful conversion of CoNiOH to CoNiP due to the in situ decomposition of $NaH_2PO_2 \cdot H_2O$ in the presence of Co and Ni ions; the standard cards JCPDS 80-0377 and JCPDS 13-0213, on the other hand, were in perfect agreement with the peak of CoNiPS@CF, indicating the successful P, S double doping of the electrode under a sulfide atmosphere and the successful preparation of the hollow nanotube array CoNiPS@CF electrode.

Figure 2b, Supplementary Fig. 4 and Supplementary Table 2 displayed the specific surface area (SSA) and corresponding pore size distribution (PSD) results. CoNiPS@CF exhibited a Type IV isotherm and the highest specific surface area ($92.975 \, m^2 \, g^{-1}$), which was attributed to its unique hollow nanotube with a graded microporous-mesoporous structure. The BJH pore size distribution results indicated that CoNiPS@CF had a large number of microporous structures and had a more mesoporous structure around 10 nm than other electrodes. The microporous structures could improve the salt removal capacity by providing copious adsorption active sites; while he wide openings of the mesoporous structure further promote ion transmission, thus improving the salt removal rate[42,43]. Correspondingly, the nanoneedle electrodes CoNiOH@CF ($39.525 \, m^2 \, g^{-1}$) and CoNiP@CF ($50.135 \, m^2 \, g^{-1}$) exhibited type III $N_2$ absorption isotherms, indicating that the nanoneedle electrodes had no pores or a small number of micropores.

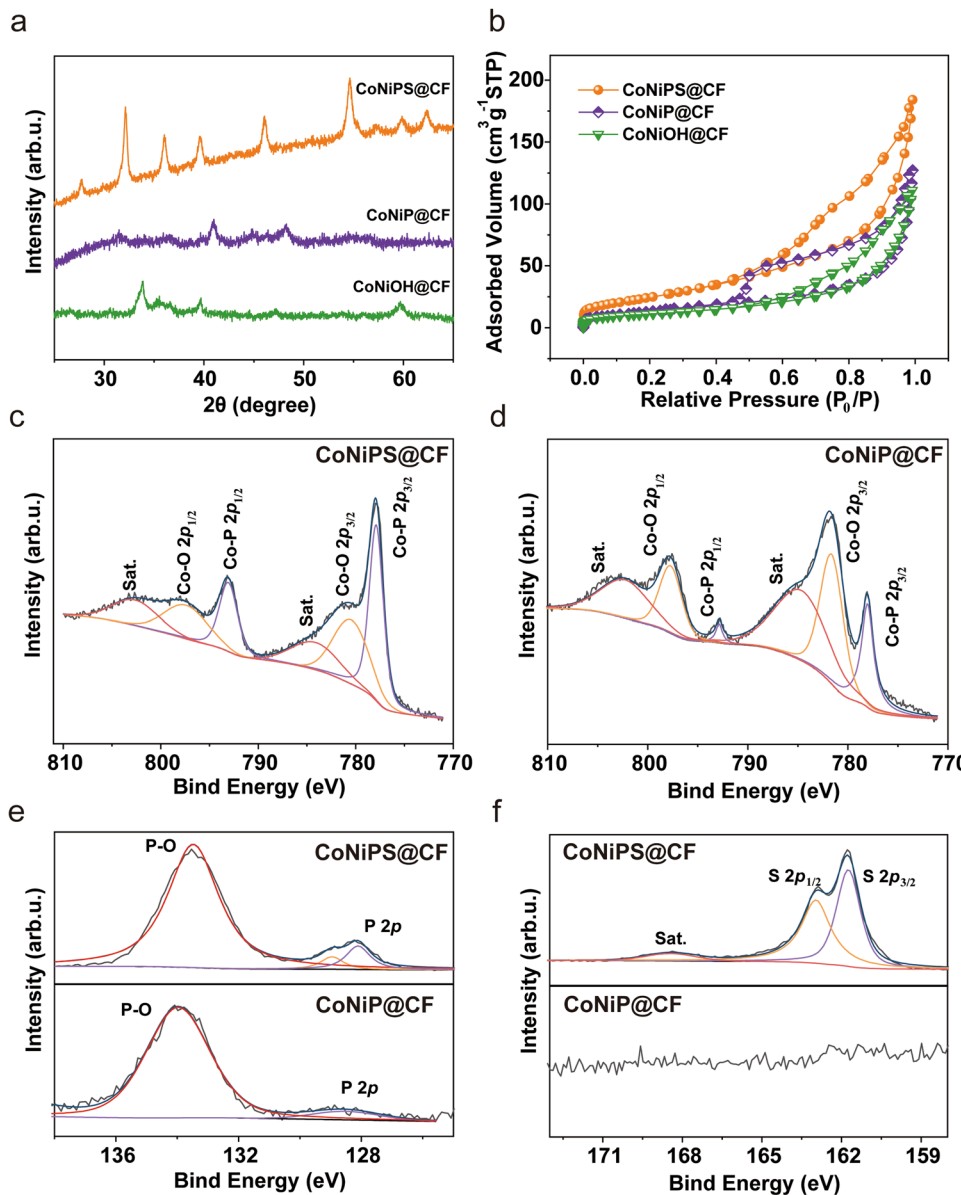

**Fig. 2 | Chemical properties of CoNiPS@CF and CoNiP@CF. a** XRD plots; **b** N$_2$ adsorption and desorption curves; XPS high-resolution of Co 2$p$ spectra of (**c**) CoNiPS@CF and (**d**) CoNiP@CF, and (**e**) P 2$p$ and (**f**) S 2$p$ for CNPS@CF and CoNiP@CF.

The chemical states of the three electrodes were studied by XPS and the total spectrum is shown in Supplementary Fig. 5. The high-resolution spectra of their different elements were analyzed. Figure 2c, d, Supplementary Fig. 6 and Supplementary Fig.7 showed the Co and Ni 2$p$ and O 1$s$ XPS spectra of CoNiPS@CF, CoNiP@CF and CoNiOH@CF. Peaks at 777.89 eV and 793.06 eV correspond to Co 2$p_{3/2}$ and Co 2$p_{1/2}$ of Co$^{3+}$ in CoNiPS@CF. The peaks at 778.03 eV and 793.29 eV correspond to Co 2$p_{3/2}$ and Co 2$p_{1/2}$ of Co$^{3+}$ in CoNiP@CF, and the two satellite peaks of CoNiPS@CF at 784.12 eV and 802.63 eV, which were assigned to the oxidized Co species[44]. Notably, the addition of sulfur to the CoNiP@CF structure resulted in a change in binding energy and an increase in partially positively charged Co, which is attributed to the bonding of sulfur to Co and form Co/P/S. The oxidation states of P in CoNiP@CF and CoNiPS@CF were shown in Fig. 2e. The peaks at 128.87 eV and 128.52 eV correspond to P$^{δ-}$ in the form of CoNiP@CF and CoNiPS@CF, while the peak around 134.1 eV is attributed to phosphate material which is produced due to the oxidation of air[45]. In addition, the P to PO ratio changed from 6.8% to 13.3% after sulfur doping, indicating that the CoNiP@CF passivation layer

was reduced because of the presence of sulfur, thereby increasing the electrode conductivity. The peaks at 161.7 eV and 162.96 eV (S 2$p_{3/2}$ and 2$p_{1/2}$, Fig. 2f) were caused by the formation of metal-S-P. The peak around 169.6 eV is sulphate species due to the surface passivation of the sulfur. It also provides the exact atomic ratios of CoNiOH, CoNiP and CoNiPS, i.e. Co:Ni = 2:1, and was close to the material dosing ratio.

## Electrochemical response and stability analysis

To further investigate the performance of the CoNiPS@CF electrode and the effect of sulfur doping, electrochemical tests were carried out using a three-electrode system in a 1 mol L$^{-1}$ NaCl solution. The CV of CoNiPS@CF presents a quasirectangular shape without observable redox peaks (Fig. 3a) indicates that CoNiPS@CF is a pseudocapacitive intercalation material like other transition metal oxides and hydroxides corroborated by galvanostatic charge-discharge (GCD) profiles without plateau (Supplementary Fig. 8); and the CV curves of the CoNiOH@CF and CoNiP@CF electrode were similar to those of CoNiPS@CF (Fig. 3b, 25 mV s$^{-1}$), indicating that the storage mechanism of Cl$^-$ for both electrodes can be attributed to pseudocapacitance[46]. The calculated values

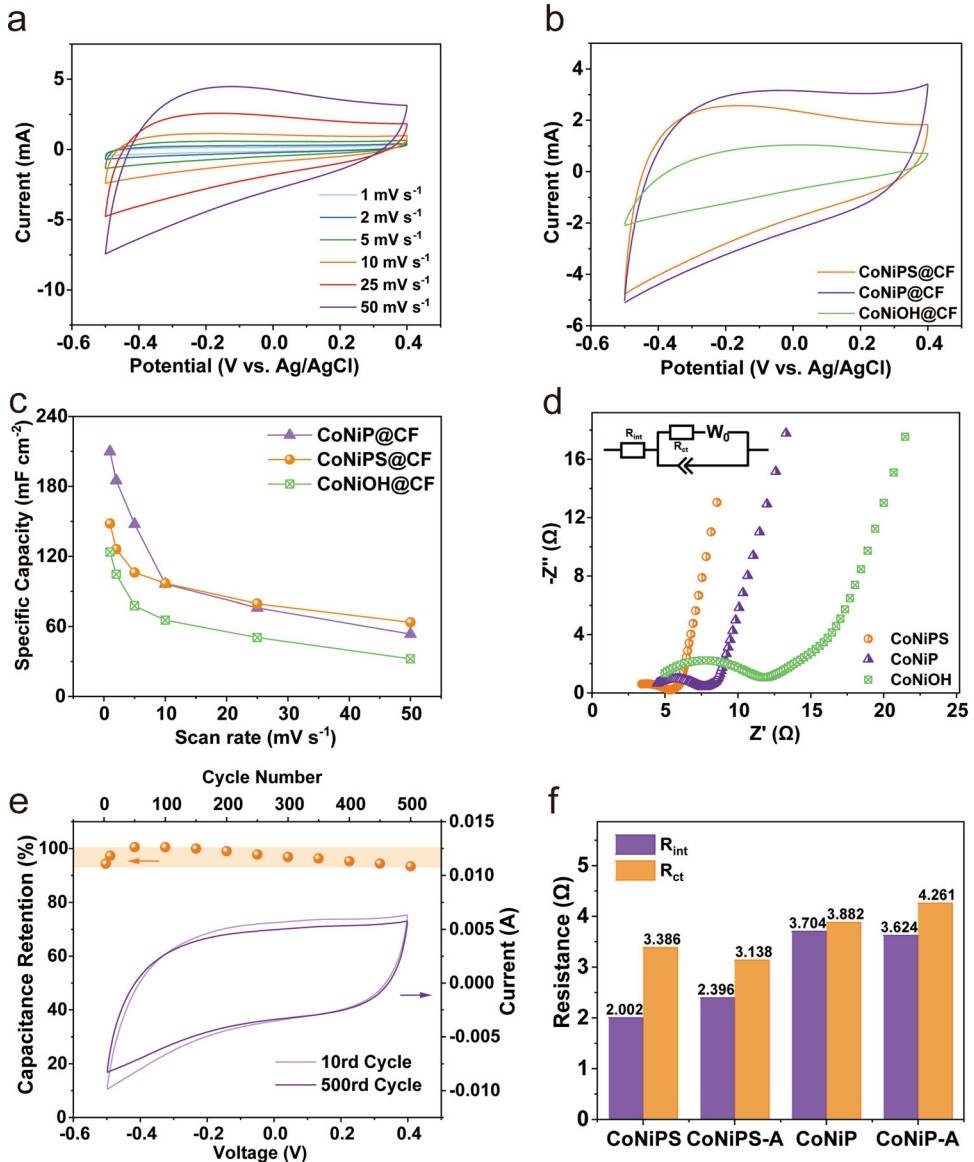

**Fig. 3 | Comparison of electrochemical performance of electrodes after phosphorus and sulfur doping. a** Cyclic voltammetry (CV) curves of CoNiPS@CF. **b** CV curves of CoNiOH@CF, CoNiP@CF and CoNiPS@CF and (**c**) comparison of specific capacity. **d** Nyquist plots of the electrodes. **e** Comparison of CV curves and capacity retention before and after 500 cycles of CoNiPS@CF and (**f**) simulated internal and charge transfer resistance values. The orange region in **e** represents the capacitance retention range.

of capacitance at different sweep rates were shown in Fig. 3c. At low sweep rates, the CoNiP@CF electrode had a higher capacitance compared to the CoNiPS@CF electrode (e.g. 209.86 mF cm$^{-2}$ at 1 mV s$^{-1}$); however, in the process of increasing the sweep speed, CoNiPS@CF can retain 42.95% of its original capacity, while CoNiP@CF retains only 25.50%. This indicates that, in comparison to CoNiP@CF, CoNiPS@CF exhibits greater stability in the electrolyte, showing reduced susceptibility to oxidation during electrochemical tests. Additionally, its specific capacitance value experiences a slower decline as the scanning rate increases, suggesting improved performance retention under varying conditions[47]. Furthermore, this phenomenon is closely associated with the distinctive hollow nanotube structure of CoNiPS@CF. This structure implies that the electrode possesses a highly open pore configuration with an abundance of accessible active sites and diminished ion diffusion kinetic constraints. CoNiPS@CF also demonstrated higher conductivity compared to CoNiP@CF, as shown by the Nyquist plot in Fig. 3d and Supplementary Table 3. In detail, compared to CoNiP@CF

(3.704 Ω) and CoNiOH@CF (4.722 Ω), CoNiPS@CF showed a lower internal resistance (R$_{int}$) value (3.082 Ω), indicating a lower percentage of charge consumption and higher charging efficiency. The charge transfer resistance (R$_{ct}$) value of the CoNiPS@CF electrode (3.386 Ω) was also lower than that of CoNiP@CF (3.882 Ω) and CoNiOH@CF (7.666 Ω), implying that CoNiPS@CF had better conductivity and superior electrochemical kinetics[48]. After the introduction of sulfur into CoNiP@CF, the ratio of P to P–O increases, resulting in a decrease in the surface passivation layer of CoNiP and an increase in the metallicity of the electrode, thereby increasing the conductivity of the electrode.

To investigate the cycling stability, a long cycle test of the CV was carried out and the results were shown in Fig. 3e. At 50 mV s$^{-1}$, the CV curve showed no significant change after 500 cycles, and the electrode specific capacity increased in the first few dozen cycles, which indicated the initial activation process of the electrode, and then there was a slight decay in the electrode specific capacity. The retentions of

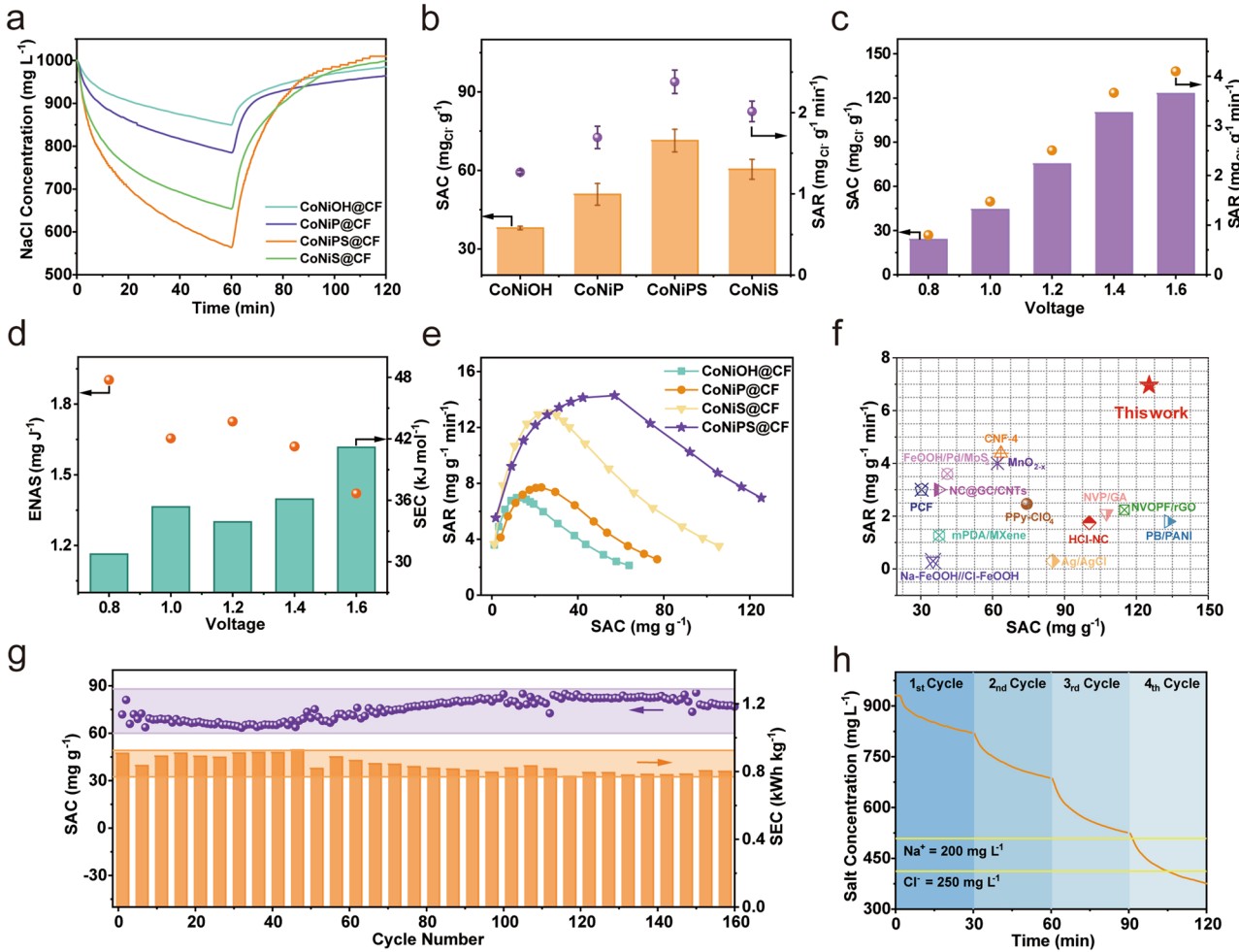

**Fig. 4 | Effect of sulfur doping on desalination performance of electrodes. a** The change of NaCl concentration versus time during CDI process and (**b**) SAC and SAR of different electrodes. An activated carbon electrode served as the cathode, and the prepared electrodes served as the anode with the area of 3 cm × 3 cm. The initial concentration of NaCl solution was 1000 mg L⁻¹. The applied voltage is 1.2 V and the time of adsorption process is 1 h, and the flow rate is 20 mL min⁻¹. Error bars are means ± standard deviation (*n* = 4 replicates). **c** SAC and SAR of CoNiPS@CF at different voltages (**d**) and corresponding ENAS and SEC. **e** Ragone plots of different electrodes. **f** Comparison of SAC and SAR of CoNiPS@CF with other state-of-the-art materials[14,42,51–67]. **g** SAC and SEC performance for 160 cycles at CoNiPS@CF in 1.0 V. The purple and orange highlighted regions represent the range of SAC and SEC respectively. **h** Plot of salt concentration change under multiple cycles in brackish water experiment.

specific capacities for the electrode were ca. 93.4% after 500 adsorption/desorption cycles measured. The high level of capacitance retention was also associated with the hollow structure of the nanotube arrays and the reduced passivation layer, which can effectively mitigate the bulk phase expansion caused by ion intercalation and significantly enhance electrode stability[49,50]. The Supplementary Fig. 9b and 9c reveals that the distinctive hollow nanotube structure of CoNiPS@CF prevents significant structural cracking throughout the cycling process, while a slight increase in the nanotube diameter is observed. In contrast, the relatively low retention of CoNiP@CF could be attributed to the formation of a oxy-hydroxide phase on the CoNiP surface[45]. Furthermore, the internal resistance and charge transfer resistance both increase with increasing number of cycles, but the resistance of the CoNiPS@CF electrode was consistently lower than that of CoNiP@CF, again demonstrating the superior electrochemical stability of CoNiPS@CF.

## Cl⁻ ion adsorption performance based on electrochemical desalination

In CDI system, to investigate the effect of the P-doping process on the electrodes, the desalination performance of an electrode not doped with phosphorus but directly doped with sulfur (noted as CoNiS@CF) was also investigated. Figure 4a illustrated the variation of the concentration of each solution in a single cycle under constant voltage conditions and Supplementary Fig. 10 showed the conductivity and current versus time images. During the charging process, the concentration of NaCl in the solution decreases rapidly, while the reversible reaction occurs during the discharging process, indicating that the electrodes have similar adsorption and desorption capabilities for Cl⁻. At a specific voltage of 1.2 V, as shown in Fig. 4b, the specific adsorption capacity (SAC) of the CoNiPS@CF electrode is as high as 71.4 ± 4.3 $mg_{Cl^-}$ g⁻¹, which was the highest desalination capacity of the individual electrodes and corresponded to the best capacitive performance of CoNiPS@CF. The adsorption capacity provided by carbon felt (CF) is very low and almost negligible compared to the active materials attached to its surface (Supplementary Fig. 11). Secondly, CoNiS@CF had a higher desalination capacity than CoNiOH@CF, which indicated that S-doping process contributed significantly to Cl⁻ adsorption capacity. Correspondingly, CoNiPS@CF exhibited the highest specific adsorption rate, averaging 2.38 ± 0.14 $mg_{Cl^-}$ g⁻¹ min⁻¹ over 30 min. This was attributed to the short Cl⁻ transport path and the higher ion adsorption rate facilitated by the highly open multi-hollow

nanotube structure of CoNiPS@CF, providing a large number of surface-accessible active sites compared to other electrodes[41]. In the constant voltage mode, voltage is a key factor for engineering applications as it is closely related to energy consumption and desalination efficiency. Figure 4c showed the adsorption capacity and desalination rate of the CoNiPS@CF electrode at different voltages. The SAC and SAR gradually increased with increasing voltage, with wider voltage intervals corresponding to longer charging times. As a result, more charge was stored at the electrode, allowing it to participate in the desalination process. When the voltage interval was expanded from $-0.8\,V/+0.8\,V$ to $-1.6\,V/+1.6\,V$, the desalination capacity of CoNiPS@CF increased from $24.0\,mg_{Cl^-}\,g^{-1}$ to $123.0\,mg_{Cl^-}\,g^{-1}$, and the specific adsorption rate increased from $0.80\,mg_{Cl^-}\,g^{-1}\,min^{-1}$ to $4.10\,mg_{Cl^-}\,g^{-1}\,min^{-1}$ was one of the highest SAR values.

Also, Fig. 4d examined the energy consumption at different voltages. It was calculated that CoNiPS@CF had an ultra-low energy-normalized adsorbed salt (ENAS) of $1.9\,mg_{NaCl}\,J^{-1}$ at $0.8\,V$, although it has a similar ENAS of approximately $1.7\,mg_{NaCl}\,J^{-1}$ at 1.0 to 1.4 V. The energy consumption value of this electrode was still low among the cutting-edge electrodes and was therefore of wide practical application. To further investigate the desalination performance of the electrode during the full adsorption process, as shown in the Ragone plot of the CDI for all prepared electrodes (Fig. 4e), CoNiPS@CF showed the highest SAR, SAC and better desalination performance. Maximum adsorption rate reached $14.28\,mg_{NaCl}\,g^{-1}\,min^{-1}$ as well as maximum adsorption capacity reached $125.33\,mg_{NaCl}\,g^{-1}$. Compared to recently reported state-of-the-art CDI electrodes including carbonaceous and Faraday materials (Fig. 4f)[14,42,51–67], CoNiPS@CF showed the highest SAC and fastest SAR (more information could be found in Supplementary Table 4). The excellent CDI performance of CoNiPS@CF stems from the functional structure of the interconnected network of porous nanotubes with extensive surface-active sites. The porous structure of the nanotube surface provides a short Cl⁻ transport pathway and therefore has an excellent kinetic promotion to the ion adsorption.

To further investigate the prospects of CoNiPS@CF for practical applications. We investigated the long-term cycling performance of the CoNiPS@CF electrode, which was performed at 1.0 V for 160 desalination cycles. It can be seen from the results that, except that the electrode was not fully activated at the beginning of the experiment and the SAC of several cycles was low, the SAC in the subsequent dozens of cycles is very stable ($86.2\%$ ~ $105.2\%$, $-72\,mg_{NaCl}\,g^{-1}$) (Fig. 4g) with no obvious signs of degradation. In addition, the SEC was calculated to be in the range of $0.78$ – $0.92\,kWh\,kg_{NaCl}^{-1}$ over 160 cycles, significantly lower than typical reported values[52]. Compared to other carbon-metal composite electrodes, CoNiPS@CF showed high capacity and excellent stability. In addition, we explored the feasibility of the device for desalination treatment experiments on brackish water (Fig. 4h). A single $3 \times 3\,cm^2$ size CoNiPS@CF was used to treat 20 ml of brackish water at 1.2 V. The results showed that the Cl⁻ concentration in the brackish water reached the drinking water standard after the 4th cycle (approx. 100 min) and that desalination was successfully achieved. The long-term cycle stability and the excellent desalination performance of brackish water suggest that CoNiPS@CF is a promising electrode for CDI applications.

## Cl⁻ ion capture mechanism analysis

The intrinsic intercalation pseudocapacitance provides maximum synergy with the fast Cl⁻ ion transport and multiple adsorption sites via the S-doping CoNiPS nanotube array. This aligns with the open (hollow) architecture and short diffusion paths within CoNiPS@CF. Supplementary Fig. 12 explored the current contribution of the CoNiPS@CF electrode for surface- and diffusion-controlled processes (Dunn analysis). The CoNiPS@CF electrode exhibited a dominant surface-controlled, achieving a peak value of $97.94\%$ at $50\,mV\,s^{-1}$ (CoNiP@CF is $70.7\%$ at $50\,mV\cdot s^{-1}$), indicating that CoNiPS@CF has a

more perfect pseudocapacitive response. It is believed that when the electrode material size is reduced to a certain nanoscale level (usually less than 10 nm), it could exhibit capacitive behavior[68–70], which could explain the higher capacitance of CoNiPS@CF electrode: the open and hollow structure and short diffusion pathways of the hollow nanotube array nanostructures provide a large amount of reaction sites and better reaction kinetics.

To further investigate the ion adsorption kinetics, one can derive key information via analysis of the dependency of the measured current with the sweep rate[71]. As can be seen from Fig. 5a, the electrode b-values calculated for CoNiPS@CF were higher than CoNiP@CF at all voltages and all above 0.8, indicating that CoNiPS@CF had more significant pseudocapacitive properties at all voltages, which can provide superior removal capacity and lower energy consumption in fast ion adsorption/desorption[72]. Further electrochemical kinetic analysis of CNPS@CF samples can be performed by Trasatti analysis method, as shown in Fig. 5b[46]. The "internal" surface referred to areas that were difficult to access, whereas the "external" surface was mainly derived from surfaces directly exposed to ions and was not affected by the sweep (more details in Methods). $q_{s,out}$ was around $72.77\,mF\,cm^{-2}$, which is $45.8\%$ of $q_s$, a relatively very high value. At a scan rate of $1\,mV\,s^{-1}$, the specific capacitance was $93.33\%$ of $q_s$, indicating the high electrochemical utilization rate of CoNiPS@CF. Thus, the CoNiPS@CF sample provided rapid, pseudocapacitive ion removal performance, which can aid in enhancing ion removal kinetics.

To interpret the more intrinsic reasons for the enhancement of performance by S-doping, the density functional theory (DFT) was therefore used to study the Cl⁻ adsorbed energy ($E_{ads}$), and difference charge density of the different adsorption sites in the CoNiP@CF and CoNiPS@CF samples. It is reported that the adsorption capability of Cl⁻ and the charge transfer ability to Cl⁻ of surface adsorption sites of the electrode materials are highly related with the electrochemical reaction activity. Figure 5c, d and Supplementary Figs. 13 and 14 show the optimized models, adsorption energy of Cl⁻, difference charge density based on Bader charge of the CoNiP@CF and CoNiPS@CF. As it can be seen, the CoNiPS@CF shows the much more negative average $E_{ads}$ of Cl⁻ in both Ni ($-3.91\,eV$) and Co sites ($-4.58\,eV$) to those ($-1.73\,eV$ for Ni, and $-1.04\,eV$ for Co) of the CoNiP@CF and also exhibits a higher charge density at the optimal adsorption site (0.52 e for CoNiPS@CF, and 0.48 e for CoNiP@CF), indicating that the surface Ni/Co atoms of CoNiPS@CF materials excel in both adsorbing Cl⁻ ions and transferring charges to Cl⁻ ions. These characteristics could promote electrochemical reactions, thereby boosting the overall reaction activity. Therefore, in addition to the kinetic convenience provided by the open structure and short diffusion paths produced by the Kirkendall effect of CoNiPS@CF electrode, the S-doping process also significantly improves the adsorption activity of Cl⁻ ions.

In this work, a porous hollow nanotube structure was constructed by utilizing the Kirkendall effect through stepwise P-doping and S-doping of CoNiOH nanoarrays grown in situ on the pCF surface, and finally a CoNiPS@CF electrode was synthesized as a CDI anode. The porous nanotube structure creates a high-speed ion channel for Cl⁻ transport, enhancing the reaction kinetics by increasing the number of open active sites; and helps to mitigate the structural damage caused by volume expansion in the desalination cycle, thus improving cycle stability. After the introduction of sulfur into CoNiP@CF, the ratio of P to P−O increases, resulting in a decrease in the surface passivation layer of CoNiP and an increase in the metallicity of the electrode, thereby increasing the conductivity of the electrode. In addition, DFT results show that the Cl⁻ adsorption energy is significantly reduced after S doping, indicating the superior capability of adsorption of Cl⁻ and the charge transfer ability to Cl⁻ of surface Ni/Co atoms of the CoNiPS@CF materials, making the electrode have better adsorption thermodynamics. Herein, the CoNiPS@CF hollow porous nanotube electrode exhibits significant

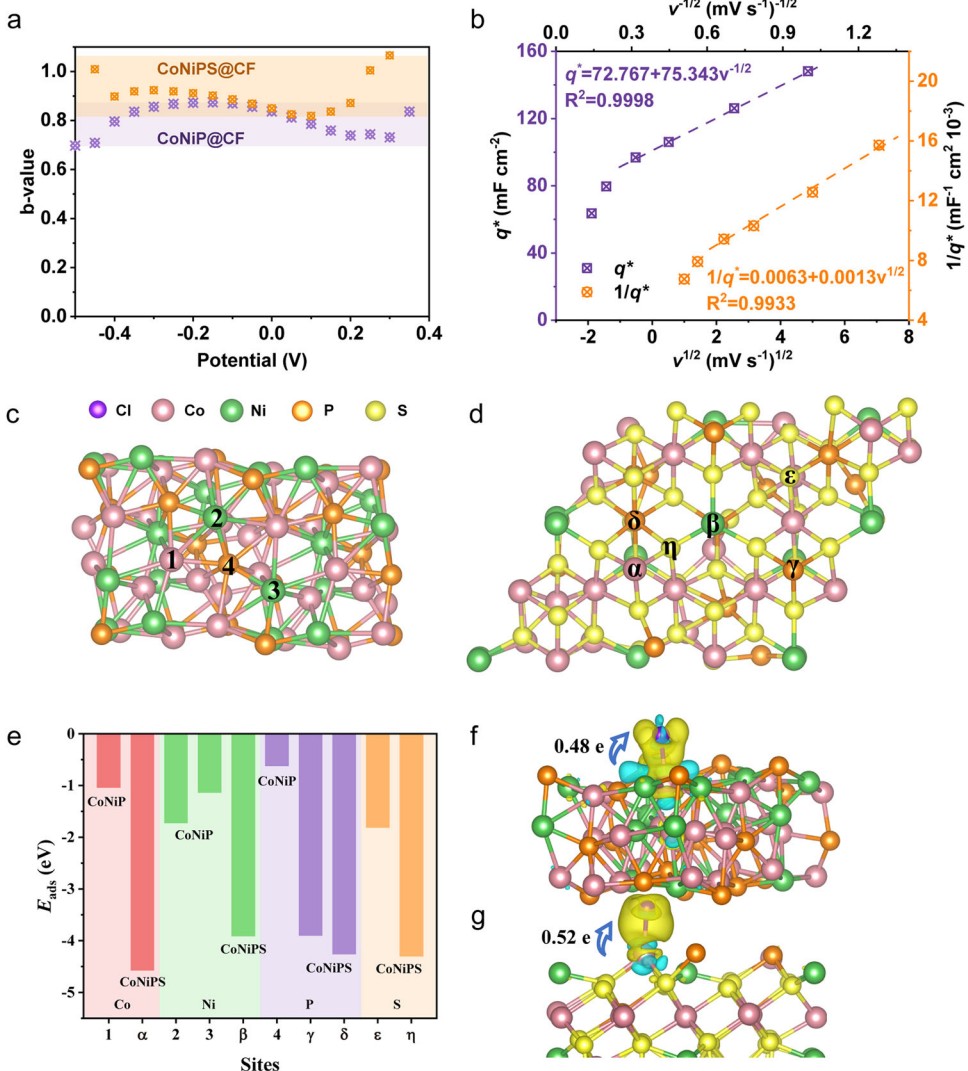

**Fig. 5 | Cl⁻ ion adsorption electrochemistry and DFT mechanism analysis.**
**a** b-values at different voltages. The purple and orange highlighted regions represent the b-value range of CoNiP@CF and CoNiPS@CF respectively. **b** The relationship between $1/q^*$ and $v^{1/2}$ and between $q^*$ and $v^{-1/2}$. The top view of the optimized (**c**) CoNiP and (**d**) CoNiPS models; **e** $E_{ads}$ of Cl⁻ at different adsorption sites; The difference charge density based on Bader charge for selected (**f**) Ni site of CoNiP and (**g**) Co site of CoNiPS to Cl⁻. Yellow represents gain of electrons and blue represents loss of electrons. The isosurface is 0.003 au.

pseudocapacitance properties, with 98% pseudocapacitance at a sweep rate of $50\,mV\,s^{-1}$ and 45.8% of the surface capacitance being 'external'. The desalination capacity of the CoNiPS@CF electrode was $76.1\,mg_{Cl^-}\,g^{-1}$ under optimal operating conditions; low energy consumption of $0.78 - 0.92\,kWh\,kg_{NaCl}^{-1}$ and a desalination rate of $6.33\,mg_{NaCl}\,g^{-1}\,min^{-1}$ is achieved. Furthermore, during the long cycle desalination, the desalination capacity of the CoNiPS@CF electrode was finally maintained at $72\,mg_{NaCl}\,g^{-1}$ with a capacity retention rate of >90%, whose desalination performance exceeds that of the state-of-the-art capacitive deionization electrodes This study provides a feasible strategy for improving the reactivity and stability of transition metal phosphides as well as for developing dechlorination electrodes with high adsorption capacities.

## Methods
### Materials
All reagents and chemicals were purchased from Sinopharm Chemical Reagent Co., Ltd., and used without further purification. All solutions were prepared using deionized water. All anion and cation exchange membranes in this experiment were purchased from Hangzhou

Huamo Technology Co., Ltd., China. The information of cation and anion exchange membranes are shown in Supplementary Table 1.

### Electrode preparation
The preparation of the pretreated carbon felt was based on the production method published by Liu et al.[73]. Under water bath conditions at 80 °C, a piece of carbon felt of $3 \times 3 \times 0.1\,cm^3$ was immersed in a solution of sulfuric acid ($H_2SO_4$, 98.0%) and nitric acid ($HNO_3$, 68%) with a volume ratio of 1:3 for 3 hours. Rinse the carbon felt removed from the solution to neutral pH with deionized water and ethanol. After ultrasonic treatment in absolute ethanol for 30 minutes, the carbon felt was washed with deionized water and placed in a vacuum drying box. After drying at 60 °C for 24 h, the pretreated carbon felt (pCF) electrode was obtained.

### Materials synthesis
CoNiOH@CF was prepared by a one-step hydrothermal method as follows: 9.6 mmol urea (≥99.0%), 3.2 mmol ammonium fluoride ($NH_4F$, ≥96.0%), 3.2 mmol nickel nitrate hexahydrate ($Ni(NO_3)_2 \cdot 6H_2O$, ≥ 98.0%) and 6.4 mmol cobalt(II) nitrate hexahydrate ($Co(NO_3)_2 \cdot 6H_2O$,

≥ 98.5%) were dissolved in 70 mL deionized water. The solution was placed in a hydrothermal reactor and the pCF was immersed in it and the reaction was sealed at 130 °C for 5 hours. After the reaction was completed and cooled to room temperature, it was washed three times with deionized water and ethanol and dried in a vacuum oven at 60 °C for 24 h to obtain CoNiOH@CF.

The dried CoNiOH@CF was placed in a tube furnace and sodium hypophosphite monohydrate ($NaH_2PO_2 \cdot H_2O$, 98.0 ~ 103.0%) 125 mg/piece electrode was placed 4 cm upstream of it, heated to 300 °C at 2 °C min⁻¹ with Argon at 50 ml min⁻¹ for 2 h for the phosphorus doping process. Then it reached room temperature in an Argon atmosphere, washed three times with deionized water and ethanol, and dried in a vacuum oven at 60 °C for 24 h to obtain a CoNiP@CF electrode.

The dried CoNiP@CF was placed in a tube furnace and sulfur (S, ≥ 99.5%) 100 mg/piece electrode was placed 13 cm upstream of it and heated to 400 °C at 10 °C min⁻¹ for 1 h in a nitrogen atmosphere for the electrode sulfur doping process. After reaching normal temperature, the electrode was washed three times with deionized water and ethanol and dried in a vacuum oven at 60 °C for 24 h to obtain a CoNiPS@CF electrode.

The area of CoNiPS@CF was calculated as shown:

$$Area = 3cm \times 3cm \times 2 + 0.1cm \times 4 = 19.2cm^2 \quad (1)$$

The calculated loading capacity per unit area of CoNiPS@CF electrode is 1.41 mg cm⁻² with 26.95 mg effective load of electrode monolith.

The preparation of the electrode without P-doping was similar (CoNiS@CF), except that the precursor electrode was changed to CoNiOH@CF and other operating conditions were unchanged.

## Materials characterization

The surface morphology was characterized by scanning electron microscopy (SEM, Hitachi S-4800/EX-350, Japan). The crystal structures of all electrodes were analyzed with X-ray diffractometer (XRD D8 ADVANCE, Bruker AXS, Germany) using Cu Kα radiation (45 kV, 40 mA). X-ray photoelectron spectrometer (XPS) analysis was performed with a ThermoFisher ESCALAB 250Xi spectrometer using monochromatized Al Ka radiation. The kinetic energy scale was calibrated by setting the C 1 s binding energy to 284.8 eV. The adsorption/desorption isotherms of $N_2$ at 77 K was used to calculate the SSA and PSD using a BELSORP Max instrument (BEL). The water contact angle (POWEREACH-JC2000D2W) was used to perform the wettability test.

## Electrochemical characterization

All electrochemical measurements were conducted in a three-electrode system using a CHI 600D electrochemical workstation (Shanghai CH Instruments Co.). A platinum sheet and Ag/AgCl were adopted as the counter and reference electrodes, respectively. The prepared electrodes were directly used as working electrode. The area of the working electrode used in the electrochemical tests was 1 cm × 1 cm. CV was swept between −0.5 V and 0.4 V under certain scan rates (1–50 mV s⁻¹), and GCD was measured at the uniform voltage window with various specific currents (1–6 mA cm⁻²). EIS spectra was tested at the frequency range of 10⁵ Hz−10⁻² Hz with an amplitude of 5 mV. All of these tests were conducted in 1 M sodium chloride (NaCl, ≥ 99.8%).

A kinetic analysis can investigate the charge storage and ion removal kinetics. Specifically, one can derive key information via analysis of the dependency of the measured current with the sweep rate[74], stated in Eqs. (1) and (2).

$$i = a\nu^b \quad (2)$$

$$\log(i) = b\log(\nu) + \log(a) \quad (3)$$

The b-value indicates the mechanism of charge storage.

The percentage of surface-controlled capacitance can be quantified by comparing it to a diffusion-controlled system (0.5 for b-value) or a surface-controlled system (1.0 for b-value). the application of Eq. (3) could be useful for this evaluation, commonly known as Dunn analysis:

$$i(V) = k_1 + k_2\nu^{\frac{1}{2}} \quad (4)$$

Trasatti method distinguishes between the "inner" and "outer" of the surface-controlled capacity[75]. To be specific, "inner" surface refers to the regions of difficult accessibility, and "outer" capacity mainly comes from the surface exposed directly to ions. Specifically, the inner−capacity refers to the active sites that are difficult to access, while the outer−capacity primarily arises from the active sites directly in contact with ions. Equations (4) and (5) were used for this calculation[46]:

$$q^* = q_{s,out} + A_1\nu^{-\frac{1}{2}} \quad (5)$$

$$q^{*-1} = q_s^{-1} + A_2\nu^{\frac{1}{2}} \quad (6)$$

where $q^*$ is the voltammetric charge, $q_{s,out}$ is the outer−capacity, and $q_s$ is the total surface-controlled capacity.

## Desalination Experiments

The flow-by CDI system is composed of end plates, gaskets, cation/anion exchange membranes and a liquid chamber ($0.8 \times 4 \times 4$ cm³). The cathode is the activated carbon electrode (3 cm × 3 cm), and the prepared electrodes used as anode (3 cm × 3 cm). All experiments were performed in batch mode, and the conductivity meter (METTLER TOLEDO S230) was used to monitor the real-time conductivity. All desalination experiments were under a constant voltage with the same flow rate (20 mL/min), the initial NaCl concentration (1000 mg L⁻¹, 20 mL), while other operational parameters, including the different activated electrodes (CoNiOH@CF, CoNiP@CF, CoNiPS@CF and CoNiS@CF) and voltage (0.8, 1.0, 1.2, 1.4 or 1.6 V). The specific adsorption capacity (SAC) and the time-average specific adsorption rate (SAR) of the prepared electrode were calculated as shown below:

$$SAC = \frac{(C_0 - C_e) \times V_d}{m} \quad (7)$$

$$SAR = \frac{SAC}{t} \quad (8)$$

where $C_O$ and $C_e$ are the initial and equilibrium NaCl concentration (mg L⁻¹), $V_d$ is the total volume(L), $m$ is the CoNiPS loading (g) on one piece of pCF. The energy-normalized adsorbed salt (ENAS, $mg_{NaCl}$ J⁻¹) was calculated according to following equation:

$$ENAS = \frac{SAC \times m}{E_{in}} = \frac{SAC \times m}{U \int_0^t I dt} \quad (9)$$

where $E_{in}$, $U$, $t$, and $I$ are energy input during charging (J), applied voltage (V), charging time (s), and current (A), respectively.

The specific energy consumption (SEC, kWh $kg_{NaCl}$⁻¹) including charging and discharging process was obtained based on following formula:

$$SEC = \frac{U \int_0^{2t} I dt}{3600 \times (C_0 - C_e) \times V_d} \quad (10)$$

## Data availability

The data that supports the findings of the study are included in the main text and supplementary information files. Raw data can be obtained from the corresponding author upon request. Source data are provided with this paper.

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

## Acknowledgements

This research is supported by The National Natural Science Foundation of China (No. 22276137, No. 52170087) and National Sponsored Postdoctoral Researcher Program of China (No. GZC20231721). We thank Xiaochen Zhang and Yuecheng Xiong for their generous support of this research.

## Author contributions

S.Y.X. designed and performed the experiments. N.N.L. and Q.L. are responsible for collecting long-cycle experiments data. M.X.L. and X.R.L. helped analyze the BET and XPS data. S.Y.X. and H.X. finished the DFT calculations. F.Y. and J.M. coordinated and supervised the research. All authors contributed to the writing of the manuscript.

## Competing interests

The authors declare no competing interests.
