## [Peer Review File · Nature Communications]

Reactive P and S co-doped porous hollow nanotube arrays for high performance chloride ion storageREVIEWER COMMENTS

Reviewer #1 (Remarks to the Author):

In this study, the importance of developing a high-performance chloride-ion storage electrode for energy recovery and water purification is emphasized. The authors developed a CoNiPS@CF electrode, which is P and S co-doped and features a porous hollow nanotube structure. The research aims to improve the reactivity and stability of transition metal phosphides, enhancing chloride storage efficiency for advanced dechlorination technologies. The work has a broader perspective, however, lacks the novelty (at least not description by the authors is not convincing) to be accepted in Nature Communications. Below are some more comments to improve the work.

- 1) Given the critical role that surface wettability plays in influencing desalination performance, an assessment of the hydrophilic nature of the materials in details would provide a more holistic understanding of their behavior and desalination efficiency.
- 2) A clear description of the sorption/desorption processes observed in the experiments is lacking. Providing a detailed account of these processes is crucial as it would offer readers a better understanding of the mechanisms at play and the implications of the findings.
- 3) The authors used hydrothermal methods for the synthesis of electrode materials. How was the loading capacity for the electrode calculated?
- 4) Why did the authors choose to limit their investigation to only 100 cycles in this experiment?
- 5) In the Results section, the explanation provided for Fig. 4a appears to correspond more accurately to Fig. 4b. It would be beneficial for the authors to review this part of the manuscript to ensure that the descriptions accurately match the respective figures.
- 6) In the Supporting Information, Section 4, Desalination Experiments, the authors used an activated carbon electrode as the anode, and the prepared electrodes served as the cathode. However, in the manuscript, lines 303 and 304 mention that activated carbon (AC) and the prepared material was used as the cathode and anode. Please clarify this discrepancy.
- 7) In the Supporting Information, Section 3, Electrochemical Test, the authors mentioned that Galvanostatic Charge-Discharge (GCD) was measured at a uniform voltage window with various specific currents. However, the GCD results are not discussed in the main manuscript. Please include the GCD results and calculate specific capacitance from GCD results.
- 8) Line 32: "electrode the to improve the efficiency". Correct the sentence.
- 9) Authors are advised to use en-dash instead of minus sign while writing units, for example g–1.
- 10) In reference section, at several places the references are not properly abbreviated.
- 11) Provide the specifications for the cation and anion exchange membranes authors used; it would be more helpful for the reader.
- 12) Correct the sample names inside the captions for Figure 2 (c) and (d)."
- 13) Incorporating the abbreviation for Carbon felt (CF) in the introduction and ensuring a clear mention of its usage can provide clarity to readers from the start.
- 14) Authors should rewrite the caption under Fig. 4a including specific details about the experimental conditions, such as the voltage applied during the test.
- 15) In the Supporting Information, in the experimental section on page 3, lines 41 to 47, there is an error

in the synthesis procedure mentioned by the authors. The correct synthesis procedure should be 'CoNiPS@CF' instead of 'CoNiP@CF.'"

Reviewer #2 (Remarks to the Author):

In this paper, porous hollow nanotube array electrode with high-speed diffusion channels and highly active sites for chloride ion adsorption was designed through stepwise phosphorus and sulfur doping based on Kirkendall effect, which promises to revolutionize the performance of energy storage and conversion systems. P doping, which enhances TMOs conductivity and electrochemical activity, poses a challenge by increasing embrittlement and blocking reactive sites. However, the author innovatively entered S doping as an ingenious solution, creating cavity and surface pore structures that reduce passivation layers. This, in turn, significantly strengthens the adsorption sites for chloride ions on the electrode's surface, effectively mitigating expansion during charging and discharging. I think this work is well constructed with high novelty but a minor revision with considering below comments is required for the possible publication in this journal.

1. The electrode material is based on carbon felt, with nanoarrays grown in situ on its surface and formed through a series of doping. In the article, the role of carbon felt does not seem to be mentioned much in the text, nor is it shown in the desalination performance, and the mass of carbon felt does not seem to be included in the calculation of active materials for SAC values. Is this reasonable? What is the desalination contribution of the original carbon material after the metal-based material is loaded? In this way, the desalination performance of the target material can be expressed more clearly.

2. It can be seen from the SEM image that the diameter of the nanoarray changes significantly before and after phosphating. The article only explains the different diameter ranges of nanoneedles/nanotubes but does not provide direct data to prove it. Additional relevant data or additional experiments are needed for further explanation. In addition, it is still necessary to more clearly analyze the impact on the physical properties of nanoneedles during the phosphorus doping and sulfur doping steps, such as why the phosphating process affects the mechanical properties of nanoneedles.

3. One of the main innovations of this article is the construction of a hollow multi-hollow nanotube array structure, which can significantly improve the chloride ion adsorption sites and cycle stability. However, in terms of cycle stability, although it showed stable performance in the desalination cycle test, the electrode structure and surface chemical properties before and after the test are still required to be characterized. At the same time, in electrochemical tests, there are also long cycle tests, but the article only compares the EIS fitting resistance changes before and after cycles. After the cycle is completed, whether there is damage to the surface structure, material crystallinity, etc. also needs to be supplemented and verified.

4. The desalination performance presented by the authors is excellent, which is very exciting! However, because CoNi-based electrodes are also one of the commonly used materials in the fields of batteries and electrocatalysis, and the materials usually have low overpotentials, it is necessary to further clarify whether there is electrolysis of water, electrocatalysis and other processes during the reaction! This is a very important factor affecting the adsorption performance of electrode materials with pseudocapacitance and EDLs as the main mechanisms.

5. Regarding the P and S doping process, as the author mentioned, there are significant differences in the physical properties such as electronegativity and diameter of the elements, so consider discussing the potential effects of P,S-doping on surface charge. If doping alters the surface charge to a negative state, it could have a significant impact on desalination performance, possibly leading to inverted CDI.. In

addition, due to the new doping of different non-metal atoms, the covalent character of the metal/non-metal bond will be affected. Does the DFT can predict the property changes for Cl ion removal? How? Please explore this aspect further in your manuscript.

6. It is essential to enhance the resolution of the figures throughout the manuscript. Improved figure quality will facilitate a better understanding of the results, which are currently challenging to interpret.

7. In the BET characterization, the authors mention that "the pore size distribution results of BJH show that CoNiPS@CF has a large number of microporous structures and more mesoporous structures around 10 nm than the other two electrodes, which suggests the presence of a hollow structure". However, there is no clear correlation between the presence of mesoporous structures and the hollow structure of the material. Therefore, please provide further explanation for the existence of a hollow structure.

8. Please further explain the dechlorination mechanism of the material.

Reviewer #3 (Remarks to the Author):

The author of this article focuses on the design of chloride ion adsorption electrodes for CDI, which is a topic in urgent need of extensive research. An innovative breakthrough in this paper is to use carbon materials as conductive substrates and dope transition metal-based materials with P and S to construct a multi-hole hollow nanotube array electrodes with a multitude of highly active sites, high-speed diffusion channels and long-term cycle stability. The excellent properties of the materials introduced in this article are so impressive and this electrode proposes excellent design solutions to improve the many bottlenecks of Cl ion storage performance and mechanical properties, so it may have significant application prospects in the future. The topic is generally interesting to the potential readers in this journal but there are some issues need to be addressed to fully support the claims in this article. Thus, a required minor revision is recommended before consideration for publication.

1.This article used a static CDI electrode, and ion exchange membranes separated the saline chamber and the electrodes. For carbon felt base materials, they are often used in flow-through configurations. I don't understand how to calculate the desalination capacity of active materials loaded on carbon felt fibers in flow-by mode? How to calculate the material loading of transition metal-based materials in carbon felt? In data presentation, would it be more appropriate to use volumetric desalting capacity?

2.After P doping, according to the author's description, it will not have a significant impact on the microstructure of the electrode surface. According to Figure S2, the CoNiP@CF sample has more microporous structure. The reason for this doesn't seem to be discussed in the manuscript. In addition, the presence of the CoNiS@CF sample in the desalination test is very reasonable, but why was the N₂ adsorption/desorption isotherms and specific surface area not tested for this sample? This can further illustrate the influence of S doping on its micromorphology.

3.The author has to show the cycled SAC-time profiles and the corresponding current-time curves for various samples. For most transition metal materials, there may be desorption difficulties in the electrode, and this irreversible adsorption has a negative impact on the cycle performance of the electrode. is extremely disadvantageous. The authors need to provide more data to prove this.

4.Water splitting is one of the common side reactions in the CDI process, especially at high voltage. The

test condition of the highest desalination capacity in this article reaches 1.8 V, which is a very high application value in transition metal materials because of their generally low overpotential. And at 1.8 V, the energy consumption of the electrode increases significantly, which may be one of the signs of water electrolysis. The authors need further discussion to prove that desalination can be performed at 1.8 V.

5. In electrochemical tests, why is the capacitance of the sulfur-doped sample lower than that of the nearly phosphorus-doped sample? This does not seem to fully correspond to the SAC data. Furthermore, why is the shape of the curve different between the long cycle CV data and the individual tests? Is there a difference in the current response between the two? The author needs further explanation.

6. In Figure 4g, there is a certain gap between the NaCl removal amount and the previous SAC value, but it seems that no significant difference in reaction conditions can be seen. Please explain the difference in SAC values, or add relevant experimental conditions and SAC-time data.

7. Why didn't the author compare the properties of CoNiOH samples in electrochemical tests? It remains somewhat unclear about the improvement in electrochemical performance described by the authors after doping.

Thank you for your letter and the reviewers' comments concerning our manuscript entitled “**Highly Reactive P, S Co-Doped Porous Hollow Nanotube-like Array of CoNiPS@CF for High-Speed, High-Capacity Cl⁻ Storage**” (Manuscript ID: NCOMMS-23-47197A). We are grateful to the reviewers for their very constructive, useful comments, which we have taken very seriously. We genuinely believe that they served to considerably strengthen and enhance the manuscript, the current revised portions are highlighted in green in the paper. Here we present our revised manuscript which has addressed all the comments and suggestions from the reviewers and editor.

Reviewer #1 (Remarks to the Author):

In this study, the importance of developing a high-performance chloride-ion storage electrode for energy recovery and water purification is emphasized. The authors developed a CoNiPS@CF electrode, which is P and S co-doped and features a porous hollow nanotube structure. The research aims to improve the reactivity and stability of transition metal phosphides, enhancing chloride storage efficiency for advanced dechlorination technologies. The work has a broader perspective, however, lacks the novelty (at least not description by the authors is not convincing) to be accepted in Nature Communications. Below are some more comments to improve the work.

Answer: We sincerely appreciate the insightful comments provided by the reviewers. To clearly highlight the innovation of this paper, we summarize the main advancements of our work as follows:

Firstly, from 2011 to the present, there has been a rapid growth in articles related to capacitive deionization (CDI) research. However, only approximately 10% of these articles focus on anode design (Acta Phys.-Chim. Sin 38 (2022): 20-31), and existing electrodes designed for chloride ion removal still face various challenges. Consequently, there is an urgent need to develop new electrodes for chlorine removal in capacitive deionization technology to address these challenges and improve overall desalination performance. In this paper, we gradually doped phosphorus and sulfur on the basis of transition metal hydroxides and designed a new method to obtain a new chloride ion removal electrode with high desalination capacity (125.33 mg g⁻¹) and high adsorption rate (6.96 mg g⁻¹ min⁻¹) in 18 minutes. This excellent performance surpasses almost all current

state-of-the-art electrodes (Fig. 1a) and shows that building a high-performance chloride ion removal electrode can significantly improve the desalination capacity of CDI.

Secondly, the novel intercalation pseudocapacitance provides maximum synergy with the fast Cl^- ion transport and multiple adsorption sites via the S-doping CoNiPS nanotube array. These beneficial electrochemical properties result in a high desalination capacity and good stability. Although transition metal phosphides have high conductivity and specific capacity, most of them often suffer from poor structural stability, which result in poor rate capability and poor long-term cycling stability. CoNiOH could generate from the reaction of CoNiP nanoneedles and electrolyte and cause a rapid decrease in the specific capacitance value (*Adv. Funct. Mater.*, 25: 7530-7538.). However, after S doping, the CoNiPS sample alleviates this problem very well and exhibits higher cycle stability and even better surface adsorption activity. This is also achieved through the Kirkendall effect, where the sulfur doping process creates cavities in situ within transition metal phosphide nanoneedles, forming porous nanotube array. This hollow (open) structure provides rapid, capacitor-like charge transfer and ion removal. Furthermore, the introduction of sulfur doping substantially decreases the adsorption energy of the adsorption sites on the electrode surface. This alteration results in a heightened affinity for Cl^- ions, enabling the electrode to demonstrate a notably high capacity for Cl^- ion adsorption.

Thirdly, the electrode holds promising prospects for industrial applications. Initially, it is synthesized in situ on the surface of carbon felt, eliminating the need for costly reagents such as binders. This not only reduces expenses but also enhances the mechanical properties of the electrode. Moreover, the preparation method is straightforward and lends itself well to scalable application. Furthermore, experiments conducted in this article demonstrate that after four consecutive cycles, the 3×3 cm CoNiPS@CF electrode, owing to its remarkable adsorption capacity, can effectively treat brackish water to meet drinking water standards within 100 minutes (Fig. 1b). Consequently, there is substantial anticipation for the application of this electrode in drinking water treatment processes and pointing out a new direction for the future development of CDI chloride ion adsorption electrodes.

Fig. 1 | **a** Comparison of SAC and SAR of CoNiPS@CF with other state-of-the-art materials. **b** Plot of salt concentration change under multiple cycles in brackish water experiment.

1) Given the critical role that surface wettability plays in influencing desalination performance, an assessment of the hydrophilic nature of the materials in details would provide a more holistic understanding of their behavior and desalination efficiency.

Answer: We thank the reviewer for this comment. Hydrophilicity not only has a huge impact on the desalination process, it also significantly affects whether the material can successfully grow in situ on the surface of the carbon felt current collector. In this case, we performed contact angle tests on all materials involved throughout the process. Before the pretreatment of the carbon felt, the hydrophilicity was extremely poor, and the water contact angle was 148.2 ° (Fig. 2a), indicating that the raw carbon felt was a hydrophobic material, which was very unfavorable for desalination and subsequent in-situ growth of the material. After being loaded with metal oxides and doped with phosphorus and sulfur, the samples all showed excellent hydrophilicity, which also changed the surface functional groups of the CF to improve its hydrophilicity. The surface of all samples is hydrophilic, and the water droplet is immersed in the surface of the carbon felt the moment it contacts the surface, and the water contact angle was 0 ° (Fig. 2b and c). This result is consistent with the desalination results. As can be seen from the Fig. 1d, Carbon Felt electrode has poor desalination performance (3.4 mgCl⁻ g⁻¹) due to its extremely strong hydrophobicity. Improving the hydrophilicity of pCF led to a notable enhancement in desalination performance (5.4 mgCl⁻ g⁻¹). However, the CoNiP@CF and CoNiPS@CF electrodes demonstrated significantly superior

desalination capabilities (50.9 and $76.1 \text{ mg}_{\text{Cl}^-} \text{ g}^{-1}$, respectively, Fig. 2d). Consequently, in this specific experiment, the material loaded onto the electrode surface emerged as the primary factor influencing its desalination performance.

Fig. 2 | The optical image of water contact angle on the surfaces of (a) CF, (b) pCF and (c) CoNiPS@CF. **d** SAC comparison between CF, pCF, CoNiP@CF and CoNiPS@CF at 1.2V (1000 mg L^{-1} NaCl solution).

2) A clear description of the sorption/desorption processes observed in the experiments is lacking. Providing a detailed account of these processes is crucial as it would offer readers a better understanding of the mechanisms at play and the implications of the findings.

Answer: We thank the reviewer for this comment. There are four main types of mechanisms for the adsorption of Cl^- by anode materials in CDI, namely electric double layer physical adsorption, conversion reaction, ion intercalation and surface redox. Different Cl^- adsorption mechanisms will show different results during the electrochemical testing process. Carbon electrode is mainly adsorbed through the electric double layer on the surface of the material, which is an electric field-assisted physical adsorption. The cyclic voltammogram (CV) of carbon electrode presents a quasirectangular shape without observable redox peaks. However, the adsorption capacity of

carbon electrodes for Cl^- is low, which is far from meeting the needs of seawater desalination. The conversion reaction refers to the material's ability to react with Cl^- to form new phases, including Ag/AgCl and Bi/BiOCl . Its CV curve will show a very prominent redox peak, and its GCD curve will have an obvious platform. Both types of electrodes have the potential to efficiently treat brine, but Ag/AgCl is expensive and the converted AgCl has poor conductivity, resulting in a slow chloride ion removal rate; Bi/BiOCl has a large volume change during the conversion process, and its cycle stability is a great challenge. The ion intercalation mechanism comes from two-dimensional materials with special structures, which can store ions between adjacent layers. LDH can accommodate anions between layers due to the positive electricity of the layers. The CV curve shows an intercalation pseudocapacitance type. In accordance with distinct reaction mechanisms, ion intercalation can be categorized into two main types: capacitor-like and battery-like mechanisms. In the capacitor-like mechanism, the CV curve appears quasirectangular and lacks discernible redox peaks. Conversely, the battery-like mechanism exhibits clear redox peaks in the CV curve, presenting an opposite profile. LDH has good potential to remove anions in capacitive deionization, but LDH has poor conductivity, so it is generally necessary to combine carbon materials or conductive polymers with better conductivity.

Fig. 3 | **a** b -values at different voltages. Normalized contribution ratios of surface-/diffusion-controlled capacities of **(b)** CoNiPS@CF and **(c)** CoNiP@CF. **d** The relationship between $1/q^*$ and $v^{1/2}$ and between q^* and $v^{1/2}$.

The CV of CoNiPS@CF presents a quasirectangular shape without observable redox peaks indicates that CoNiPS@CF is a pseudocapacitive intercalation material like other transition metal oxides and hydroxides corroborated by GCD profiles without plateau. Besides, the nanotube array electrode is prone to be pseudocapacitive as a result of free ion diffusion (Chem. Rev. 2020, 120, 14, 6738–6782). This aligns with the open (hollow) architecture and short diffusion paths within CoNiPS@CF. A kinetic analysis can further investigate the charge storage and ion removal kinetics. Specifically, one can derive key information via analysis of the dependency of the measured current with the sweep rate.

$$i = av^b$$

$$\log(i) = b\log(v) + \log(a)$$

The b -value can be determined from the plot's slope between $\log(i)$ and $\log(v)$ and indicates the charge storage mechanism. As b -values approach 0.5, the current followed a diffusion-controlled law, which was typically seen in battery-like systems. In contrast, a b -value closer to 1.0 indicated the ideal surface-controlled situation, typically seen in capacitor-like systems. As can be seen from Fig. 3a, the electrode b -values calculated for CoNiPS@CF were higher than CoNiP@CF at all voltages and all above 0.8, aligning with the pronounced pseudocapacitive behavior of the electrode material, which can provide superior rate capability and lower energy consumption in fast ion adsorption/desorption. It is also possible to quantify the percentage of surface-controlled capacitance (capacitor-like contribution) corresponding with either a perfect diffusion-limited system (b -value of 0.5) or a perfect capacitor (b -value of 1.0). The closer it is to one, the more perfect is the pseudocapacitive response. A beneficial calculation for this consideration often referred to as Dunn analysis:

$$i(V) = k_1v + k_2v^{1/2}$$

In this equation, k_1 corresponds with an ideal (pseudo)capacitive contribution and k_2 with a battery-like feature. For CoNiPS@CF material, k_1 represents 98% of the total capacity at the scan rate of $50 \text{ mV}\cdot\text{s}^{-1}$ (Fig. 3b). As a comparison, this percentage for CoNiP@CF is 70.7% at $50 \text{ mV}\cdot\text{s}^{-1}$ (Fig. 3c). The Trasatti analytical method was used to further analyze the electrochemical kinetics of

CNPS@CF samples as shown in Fig. 3d. This method distinguished the surface control capability of the CoNiPS@CF electrode into 'internal' and 'external' surface control. The "internal" surface referred to areas that were difficult to access, whereas the "external" surface was mainly derived from surfaces directly exposed to ions and was not affected by the sweep.

$$q^* = q_{s,out} + A_1 v^{-1/2}$$

$$q^{*-1} = q_s^{-1} + A_2 v^{1/2}$$

$q_{s,out}$ was calculated to be 72.77 mF cm^{-2} , which is 45.8% of q_s , a relatively very high value. At a scan rate of 1 mV s^{-1} , the specific capacitance of CoNiPS@CF electrode ($148.24 \text{ mF cm}^{-2}$) was 93.33% of q_s , indicating the outstanding electrochemical utilization rate of CoNiPS@CF. Thus, the CoNiPS@CF electrode provided rapid, capacitor-like ion removal and charge transfer, which can be beneficial in enhancing ion storage kinetics.

Fig. 4 | The top view of the (a) CoNiP and (b) CoNiPS optimized structural models; c E_{ads} of Cl^- with different adsorption sites; The difference charge density and Bader charge analysis for selected (d) Ni site of CoNiP and (e) Co site of CoNiPS to Cl^- .

To try to interpret the more intrinsic reasons for the enhancement of performance by S-doping, the density functional theory (DFT) was therefore used to study the Cl^- adsorbed energy and difference charge density of the adsorption sites in the CoNiP and CoNiPS electrodes. It is reported that the adsorption capability of Cl^- and the charge transfer ability to Cl^- of surface adsorption sites of the electrode materials are highly related with the electrochemical reaction activity. Fig. 4 show the optimized structural models, adsorption energy of Cl^- (E_{ads}), difference charge density, and Bader charge analysis of the CoNiP@CF and CoNiPS@CF. As it can be seen, the CoNiPS@CF shows the much more negative average E_{ads} of Cl^- in both Ni (-3.91 eV) and Co sites (-4.58 eV) to those (-1.73 eV for Ni, and -1.04 eV for Co) of the CoNiP@CF and also exhibits a higher charge density at the optimal adsorption site (0.52 e for CoNiPS@CF, and 0.48 e for CoNiP@CF), indicating the superior capability of adsorption of Cl^- and the charge transfer ability to Cl^- of surface Ni/Co atoms of the CoNiPS@CF materials, which may facilitate the electrochemical reactions and thus enhance the reaction activity. Therefore, in addition to the kinetic convenience provided by the open structure and short diffusion paths produced by the Kirkendall effect of CoNiPS@CF electrode, the S-doping process also significantly improves the adsorption activity of Cl^- ions.

3) The authors used hydrothermal methods for the synthesis of electrode materials. How was the loading capacity for the electrode calculated?

Answer: We thank the reviewer for this comment. We obtain the load mass by subtracting the mass of the carbon felt (CF) from the mass of the final sample. Specifically, weigh the dried CF to obtain the mass of CF. And after the hydrothermal process is completed, the sample is washed by water and ethanol and dried in vacuum oven, and the dried sample is weighed to obtain the total mass of the final sample (like CoNiPS@CF). Therefore, the load mass of CoNiPS can be obtained by subtracting the two masses and the loading capacity is obtained by dividing the load mass of

CoNiPS by the apparent area of the single CF ($3\text{ cm} \times 3\text{ cm} \times 0.1\text{ cm}$, the area is $3 \times 3 \times 2 + 3 \times 0.1 \times 4 = 19.2\text{ cm}^2$).

The removal capacity of NaCl based on the prepared electrode area were calculated as shown below.

$$\text{desalination capacity}(Q) = \frac{(C_0 - C_e)V}{M} \text{ (mg of NaCl/g of CoNiPS@CF)} \quad (4)$$

where C_0 and C_e are the NaCl concentration (mg L^{-1}) at initial and equilibrium stages, respectively, V is the total NaCl volume(L), M is the loading mass of CoNiPS in one piece of CF.

In order to have a more specific expression, we have added this description to Supplementary Section 1 and Section 4.

Supplementary Page 3:

The area of CoNiPS@CF was calculated as shown:

$$\text{Area} = 3\text{ cm} \times 3\text{ cm} \times 2 + 3\text{ cm} \times 0.1\text{ cm} \times 4 = 19.2\text{ cm}^2$$

So the calculated loading capacity per unit area of CoNiPS@CF electrode is 1.41 mg cm^{-2} , and the effective load of electrode monolith is 26.95 mg.

Supplementary Page 7:

where C_0 and C_e are the NaCl concentration (mg L^{-1}) at initial and equilibrium stages, respectively, V_d is the total NaCl volume(L), M is the mass of CoNiPS loading on one piece of CF,

4) Why did the authors choose to limit their investigation to only 100 cycles in this experiment?

Answer: We thank the reviewer for this comment. The end of a long-term experiment is marked by the Cell shutdown; but for the desalination process, waiting for the cell to shut down is difficult. Therefore, when the desalination capacity is lower than 80% of the initial performance, the long cycle experiment can be regarded as terminated (Research, 2021. DOI:10.34133/2021/9754145). From the results of our long cycle test, we can see that the desalination capacity is very stable and there is no significant decrease in capacity after 100 cycles, which aligns with the open (hollow)

architecture and short diffusion paths within CoNiPS@CF. The SAC of CoNiPS@CF showed an excellent reversibility of 91.2% ~ 103.8% and was consistently maintained at 72 mg_{NaCl} g⁻¹ (1000 mg L⁻¹ NaCl solution, at 1.0 V constant voltage for 1 h). This performance ranks nearly at the pinnacle among the transition metal-based CDI state-of-the-art materials, considering both cycle stability and desalination capacity. Therefore, due to limitations of laboratory conditions, we limited the cycles to 100 and the observed performance in transition metal-based electrodes is quite high.

In order to further demonstrate that the material has excellent cycle stability, we extended the number of cycles of the desalination experiment to 160 cycles (Fig. 5a). It can be seen from the results that, except that the electrode was not fully activated at the beginning of the experiment and the SAC of several cycles was low, the SAC in the subsequent dozens of cycles is very stable (86.2% ~ 105.2%), which is in the same trend with our previous 100-cycle test. However, after 160 desalination cycles, the brine has continued to flow for nearly 320 h. During this process, the solution will continue to evaporate, so if the experiment is continued, the accuracy of the results will be affected.

To better verify the reversibility under long cycle conditions, also conducted cyclic voltammetry (CV) tests for 500 cycles under 50 mV/s (Fig. 5b). In long-cycle CV tests, similar to the desalination experiment, the electrode specific capacity increased in the first few dozen cycles, which indicated the initial activation process of the electrode, and then there was a slight decay in the electrode specific capacity. The cyclic voltammograms' shapes also display no significant change after 500 cycles. The retentions of specific capacities for the electrode were ca. 93.4 % after 500 adsorption/desorption cycles measured, which also proves that its electrochemical reversibility is very outstanding.

Before and after the long CV cycling, we also explored the changes in the morphology of its nanotube arrays. The Fig. 5c and 5d reveals that the surface nanotube array morphology of the electrode remains unchanged, while a slight increase in the nanotube diameter is observed. This change is attributed to the partial bulk expansion of the electrode caused by the adsorption of chloride ions. Notably, the distinctive hollow nanotube structure of CoNiPS@CF prevents significant structural cracking throughout the cycling process. This observation underscores the crucial role of this specific structure in preserving the long-term cycling stability of the electrode.

Fig. 5 | **a** SAC and SEC performance for 160 desalination cycles at CoNiPS@CF in 1.0 V. **b** Specific capacity retention rate of 500 CV cycles at 50mV/s. Electrode morphology (**c**) before and (**d**) after 500 CV cycles.

5) In the Results section, the explanation provided for Fig. 4a appears to correspond more accurately to Fig. 4b. It would be beneficial for the authors to review this part of the manuscript to ensure that the descriptions accurately match the respective figures.

Answer: We thank the reviewer for this comment. Fig. 4a illustrated the variation of concentration of each solution during charging and discharging process, which aims to explore the adsorption and desorption process of a certain electrode. And it can also show the electrode has similar adsorption capacity and desorption capacity, indicating that no apparent side reactions occur. Fig. 4b showed the specific adsorption capacity (SAC) and specific adsorption rate (SAR) of different electrodes at 1.2V within 30 min and also represented CoNiPS@CF has best SAC and SAR among all electrodes. Therefore, we've modified the description to more accurately match the corresponding figures.

Fig. 4a illustrated the variation of the concentration of each solution in a single cycle under constant voltage conditions and Supplementary Fig. 10 showed the conductivity and current versus time images. During the charging process, the concentration of NaCl in the solution decreases rapidly, while the reversible reaction occurs during the discharging process, indicating that the electrodes have similar adsorption and desorption capabilities for Cl^- . At a specific voltage of 1.2 V, as shown in Fig. 4b, the specific adsorption capacity (SAC) of the CoNiPS@CF electrode is as high as $71.4 \pm 4.3 \text{ mgCl}^- \text{ g}^{-1}$, which was the highest desalination capacity of the individual electrodes and corresponded to the best capacitive performance of CoNiPS@CF.

6) In the Supporting Information, Section 4, Desalination Experiments, the authors used an activated carbon electrode as the anode, and the prepared electrodes served as the cathode. However, in the manuscript, lines 303 and 304 mention that activated carbon (AC) and the prepared material was used as the cathode and anode. Please clarify this discrepancy.

Answer: We thank the reviewer for this comment. In our experiment, the activated carbon was used as cathode and the prepared materials served as the anode. So the information in SI is wrong. We have corrected this information.

Supplementary Page 6:

An activated carbon electrode served as the cathode, and the prepared electrodes served as the anode with the area of $3 \text{ cm} \times 3 \text{ cm}$.

7) In the Supporting Information, Section 3, Electrochemical Test, the authors mentioned that Galvanostatic Charge-Discharge (GCD) was measured at a uniform voltage window with various specific currents. However, the GCD results are not discussed in the main manuscript. Please include the GCD results and calculate specific capacitance from GCD results.

Answer: We thank the reviewer for this comment and kind reminder. Now we add the GCD data of related materials. Galvanostatic Charge-Discharge (GCD) experiments were conducted on all samples at a current density of 1 mA cm^{-2} (Fig. 6a). Among the samples, CoNiPS@CF

demonstrated the highest specific capacitance of $0.103 \text{ mAh cm}^{-2}$, which is significantly higher than CoNiOH@CF and CoNiP@CF , with specific capacitance of $0.075 \text{ mAh cm}^{-2}$, and $0.034 \text{ mAh cm}^{-2}$, respectively. Furthermore, no plateau was observed at different specific currents (Fig. 6b), suggesting that the CoNiPS@CF samples work through a pseudocapacitive mechanism, like other transition metal oxides and hydroxides. Similar conclusions can also be drawn from cyclic voltammetry test, representing a quasirectangular shape without observable redox peaks.

Fig. 6 | **a** The galvanostatic charge-discharge (GCD) profiles of different samples at 1 mA cm^{-2} ; **b** GCD profile of CoNiPS@CF electrode at different current density.

8) Line 32: “electrode the to improve the efficiency”. Correct the sentence.

Answer: We thank the reviewer for this comment. We have corrected this sentence and checked all other sentences in the rest of the article for accuracy.

Page 2:

In this work, we present a approach to improve the reactivity and stability of transition metal phosphides as well as enhance the reaction rate and the active open site for chloride storage, devoting to improve the efficiency of advanced dechlorination technologies.

9) Authors are advised to use en-dash instead of minus sign while writing units, for example g–l.

Answer: We thank the reviewer for this comment. According to reviewer's suggestion, we have modified the format of units throughout the manuscript.

10) In reference section, at several places the references are not properly abbreviated.

Answer: We thank the reviewer for this comment. We have made all modifications in accordance with the document format requirements of *Nature Communications*.

11) Provide the specifications for the cation and anion exchange membranes authors used; it would be more helpful for the reader.

Answer: We thank the reviewer for this comment. All anion and cation exchange membranes in this experiment were purchased from Hangzhou Huamo Technology Co., Ltd., China. The information of cation and anion exchange membranes are as follows:

Table 1. Ion exchange membrane parameter information

Parameter	CEM8040	AEM8040
Switching Capacity (mol kg ⁻¹)	2.2	2.0
Selective Permeability (%) ≥	95 %	95 %
Acid-base Tolerance Concentration(mol L ⁻¹) ≤	3 mol L ⁻¹	3
Wet-state Thickness (mm)	0.30 – 0.35	0.16 – 0.23
Membrane Resistance (Ω cm ⁻² ,0.5M NaCl, 25°C) ≤	5.5	4.0
Current Density (A m ⁻²) ≤	400	300
Water Permeability (mL h ⁻¹ cm ⁻² MPa ⁻¹) ≤	0.1	0.1
Thermal Stability (°C) ≤	50	50

We have added the information of CEM and AEM to Section 4 of Supporting Information.

12) Correct the sample names inside the captions for Figure 2 (c) and (d)."

Answer: We thank the reviewer for this comment. We have checked the information of figures and correct the captions of (c) and (d). Fig. 2c and Fig. 2d are the high-resolution spectra of Co element of CoNiPS@CF and CoNiP@CF electrodes, respectively.

Page 10:

Fig. 2 | Chemical properties of CoNiPS@CF and CoNiP@CF. a XRD plots; **b** N₂ adsorption and desorption curves; XPS high-resolution of Co 2p spectra of **(c)** CoNiPS@CF and **(d)** CoNiP@CF, and **(e)** P 2p and **(f)** S 2p for CNPS@CF and CoNiP@CF.

13) Incorporating the abbreviation for Carbon felt (CF) in the introduction and ensuring a clear mention of its usage can provide clarity to readers from the start.

Answer: We thank the reviewer for this comment. We added the appropriate abbreviations where carbon felt (CF) and pretreated carbon felt (pCF) first appear. CF is a very hydrophobic material, so it is not conducive to the in-situ growth of the active material during the hydrothermal process. After pretreatment, the pCF surface becomes very hydrophilic, so CoNiOH nanowires can be grown on the surface of pCF. Besides, the in-situ growth of active materials on the surface of pCF fibers can avoid the use of binders, thereby avoiding problems such as the decrease in conductivity and cycle stability caused by the presence of binders.

Page 5:

P, S co-doped porous hollow nanotube array electrodes with high-speed diffusion channels and highly active sites is obtained by gradually phosphating and sulfuring CoNiOH nanoarrays grown in situ on the surface of pretreated carbon felt (pCF) and is used as the anode in HCDI for Cl⁻ removal.

Page 16:

The adsorption capacity provided by carbon felt (CF) is very low and almost negligible compared to the active materials attached to its surface (Supplementary Fig. 11).

14) Authors should rewrite the caption under Fig. 4a including specific details about the experimental conditions, such as the voltage applied during the test.

Answer: We thank the reviewer for this comment. Fig. 4a and 4b represent the changes in NaCl concentration, specific adsorption capacity (SAC) and specific adsorption rate (SAR) under different electrodes during the desalination process. In this process, the initial NaCl was 1000 mg L⁻¹ with 20 mL, and the experiment was carried out under a constant voltage of 1.2V with a running time of 1h. An activated carbon electrode served as the cathode, and the prepared electrodes served as the anode with the area of 3 cm × 3 cm. Therefore, we added the special experimental conditions in this process to the caption and to the text.

Page 15:

Fig. 4 | Effect of sulfur doping on desalination performance of electrodes. a The change of NaCl concentration versus time during CDI process and **(b)** SAC and SAR of different electrodes. An activated carbon electrode served as the cathode, and the prepared electrodes served as the anode with the area of 3 cm × 3 cm. The initial concentration of NaCl solution was 1000 mg L⁻¹. The applied voltage is 1.2 V and the time of adsorption process is 1 hour, and the flow rate is 20 mL min⁻¹.

15) In the Supporting Information, in the experimental section on page 3, lines 41 to 47, there is an error in the synthesis procedure mentioned by the authors. The correct synthesis procedure should be 'CoNiPS@CF' instead of 'CoNiP@CF.'"

Answer: We thank the reviewer for this comment. We checked all preparation methods in the experimental section and corrected any erroneous information.

Supplementary Page 3:

The dried CoNiP@CF was placed in a tube furnace and sulfur 100 mg/piece electrode was placed 13 cm upstream of it and heated to 400 °C at 10 °C min⁻¹ for 1 h in a nitrogen atmosphere

for the electrode sulfur doping process. After reaching normal temperature, the electrode was washed repeatedly with deionized water and ethanol, and dried in a vacuum oven at 60°C for 24 h to obtain a CoNiPS@CF electrode.

The preparation of the electrode without P-doping was similar (CoNiS@CF), except that the precursor electrode was changed to CoNiOH@CF and other operating conditions were unchanged.

Reviewer #2 (Remarks to the Author):

In this paper, porous hollow nanotube array electrode with high-speed diffusion channels and highly active sites for chloride ion adsorption was designed through stepwise phosphorus and sulfur doping based on Kirkendall effect, which promises to revolutionize the performance of energy storage and conversion systems. P doping, which enhances TMOs conductivity and electrochemical activity, poses a challenge by increasing embrittlement and blocking reactive sites. However, the author innovatively entered S doping as an ingenious solution, creating cavity and surface pore structures that reduce passivation layers. This, in turn, significantly strengthens the adsorption sites for chloride ions on the electrode's surface, effectively mitigating expansion during charging and discharging. I think this work is well constructed with high novelty but a minor revision with considering below comments is required for the possible publication in this journal.

1. The electrode material is based on carbon felt, with nanoarrays grown in situ on its surface and formed through a series of doping. In the article, the role of carbon felt does not seem to be mentioned much in the text, nor is it shown in the desalination performance, and the mass of carbon felt does not seem to be included in the calculation of active materials for SAC values. Is this reasonable? What is the desalination contribution of the original carbon material after the metal-based material is loaded? In this way, the desalination performance of the target material can be expressed more clearly.

Answer: We thank the reviewer for this comment. the carbon felt does play a very important role in the desalination process. However, its role is more as a conductive fiber, providing a base for the attachment of active materials. The in-situ growth of active materials on the surface of CF fibers can avoid the use of binders, thereby avoiding problems such as the decrease in conductivity and cycle stability caused by the presence of binders. In fact, the presence of CF fiber does provide some desalination capacity, which was also mentioned in our previous study (Nano-Micro Lett. 16, 143 (2024)); however, the adsorption capacities of pCF and CF are only 7% and 4% of the CoNiPS@CF electrode respectively. Thus, the adsorption capacity provided by CF is very low and almost negligible compared to the active materials attached to its surface (Fig. 1).

Fig.1 | SAC comparison between CF, pCF, CoNiP@CF and CoNiPS@CF at 1.2V (1000 mg L⁻¹ NaCl solution).

2. It can be seen from the SEM image that the diameter of the nanoarray changes significantly before and after phosphating. The article only explains the different diameter ranges of nanoneedles/nanotubes but does not provide direct data to prove it. Additional relevant data or additional experiments are needed for further explanation. In addition, it is still necessary to more clearly analyze the impact on the physical properties of nanoneedles during the phosphorus doping and sulfur doping steps, such as why the phosphating process affects the mechanical properties of nanoneedles.

Answer: We thank the reviewer for this comment. From the SEM images, we can see the changes in nanoneedle diameter before and after different doping processes. We counted the diameters of the 50 nanoneedles in the SEM pictures, as shown in Fig. 2a, 2b and 2c. The statistical results are basically consistent with the normal distribution. From the results, we can see that the diameter of CoNiOH is about 85.28 ± 14.11 nm, which is slightly smaller than the diameter of CoNiP (86.73 ± 12.94 nm). This is because during the chemical vapor deposition (CVD) process, the initial transition metal hydroxide is doped with phosphorus to form the transition metal phosphide, so its diameter also changes. In addition, the advantage of the CVD method is that it has little change in the original electrode morphology and can maintain its original high specific surface area nanoneedle morphology. However, when we further doped sulfur, the surface nanoneedle

morphology changed, and porous nanotubes were gradually formed from the nanoneedles. This is due to the significant Kirkendall phenomenon in the sulfur doping process. S^{2-} first reacted with CoNiP to form a thin layer of CoNiPS, which acted as a barrier to slow down the reaction between the external S^{2-} and the internal CoNiP. Based on the non-equilibrium diffusion process between the outward CoNiP and the inward S^{2-} , voids were created in the center of the nanoneedles. As the reaction proceeds, the CoNiPS shell thickens and the CoNiP nuclei gradually decrease, eventually forming CoNiPS nanotubes. Therefore, the average diameter of CoNiPS is the largest among the three samples, which is 106.64 ± 16.6 nm (Fig. 2c). Through the TEM picture of a single sample, we can more intuitively see the difference in diameter (Fig. 2c and 2d). The average diameter of CoNiP is 83.99 nm, while the diameter of CoNiPS is 114.27 nm.

Fig.2 | Diameter distribution of (a) CoNiOH@CF, (b) CoNiP@CF nanoneedles and (c) CoNiPS@CF nanotubes from SEM. TEM images of single (d) CoNiP@CF nanoneedle and (e) CoNiPS@CF nanotube.

The hollow nanotube CoNiPS formed after sulfur doping has significant advantages over nanoneedle CoNiP. First, from the perspective of physical properties, the hollow configuration can play a cushion effect on bulk expansion during the ion intercalation process. Finite element analysis shows that this configuration can alleviate the pressure during the ion intercalation process (Research, 2021. DOI:10.34133/2021/9754145). For a solid nanoneedle, the stress exhibited at the

edge and the shear stress caused by the Cl^- ion intercalation are greater than the stress of a hollow nanotube, thus causing greater pressure. During expansion, the physical deformation of materials is expressed as displacement, that is, bias relative to the initial position. The displacement in a hollow nanotube is generally smaller. Therefore, the hollow structure is more stable as it undergoes less distortion and stress when charging. Second, finite element simulation is applied to analyze the ion concentration distribution during desalination and to demonstrate that a hollow structure is favorable for ion transport (ACS Appl. Mater. Interfaces 2020, 12, 2, 2180–2190). The superior structure of the CoNiPS nanotube substantially promotes the ion transfer rate by shortening ion diffusion paths in the cavity of the electrode material. Also, both inner and outer walls of the CoNiPS nanotube provide sufficient active sites for fast adsorption and desorption of salty ions. Third, most transition metal phosphides often suffer from poor structural stability, which result in poor rate capability and poor long-term cycling stability. CoNiOH could generate from the reaction of CoNiP nanoneedles and electrolyte and cause a rapid decrease in the specific capacitance value (Adv Funct Mater 25, 7530-7538 (2015)) However, after S doping, the CoNiPS sample alleviates this problem very well and exhibits higher cycle stability and even better surface adsorption activity. In the process of increasing the sweep speed (Fig. 3a), CoNiPS@CF can retain 42.95% of its original capacity, while CoNiP@CF retains only 25.50%. This may be because the CoNiP@CF sample is unstable and easily oxidized to produce a passivation layer, which can also be derived from the XPS analysis results: the P to PO peak ratio changed from 6.8% to 13.3% after sulfur doping (Fig. 3b), suggesting that the presence of sulfur reduced the formation of the CoNiP@CF passivation layer.

Fig. 3 | a specific capacity comparison of CoNiOH@CF, CoNiP@CF and CoNiPS@CF. **b** XPS high-resolution spectra of P 2p for CoNiPS@CF and CNP@CF.

3. One of the main innovations of this article is the construction of a hollow multi-hollow nanotube array structure, which can significantly improve the chloride ion adsorption sites and cycle stability. However, in terms of cycle stability, although it showed stable performance in the desalination cycle test, the electrode structure and surface chemical properties before and after the test are still required to be characterized. At the same time, in electrochemical tests, there are also long cycle tests, but the article only compares the EIS fitting resistance changes before and after cycles. After the cycle is completed, whether there is damage to the surface structure, material crystallinity, etc. also needs to be supplemented and verified.

Answer: We thank the reviewer for this comment. In order to further demonstrate that the material has excellent cycle stability, we extended the number of cycles of the desalination experiment to 160 cycles (Fig. 4a). It can be seen from the results that, except that the electrode was not fully activated at the beginning of the experiment and the SAC of several cycles was low, the SAC in the subsequent dozens of cycles is very stable (86.2% ~ 105.2%), which is in the same trend with our previous 100-cycle test. In order to better verify the reversibility under long cycle conditions, also conducted cyclic voltammetry (CV) tests for 500 cycles under 50 mV/s (Fig. 4b). In long-cycle CV tests, similar to the desalination experiment, the electrode specific capacity increased in the first few dozen cycles, which indicated the initial activation process of the electrode, and then there was a slight decay in the electrode specific capacity. The cyclic voltammograms' shapes also display no significant change after 500 cycles. The retentions of specific capacities for the electrode were ca. 93.4 % after 500 adsorption/desorption cycles measured, which also proves that its electrochemical reversibility is very outstanding.

From the results of Fig. 4c, it can be seen that the XRD of the sample after 500 CV cycling is still basically consistent with the original sample. But there are still some new crystal phases formed: at $2\theta = 20.16^\circ$, this is the crystal phase of $\text{CoO}(\text{OH})$, which represents the oxidation phenomenon caused by the partial reaction between the surface electrode and the electrolyte during the adsorption process; at $2\theta = 21.98^\circ$ and 35.97° , it shows that the adsorption and desorption of ions on the electrode surface leads to the destruction of the original crystal phase and the formation of new crystal phases ($\text{P}_4\text{O}_6\text{S}_2$ and $\text{Ni}_2\text{P}_2\text{S}_6$). However, even with the presence of these new crystal

phases and a certain degree of oxidation, there was no obvious decline for desalination performance. So the high retention of electrochemical capacities for the electrode contributed to the excellent cycling performance due to the partial corrosion to a low extent.

Before and after the long CV cycling, we also explored the changes in the morphology of its nanotube arrays. The Fig. 4d and 4e reveals that the surface nanotube array morphology of the electrode remains unchanged, while a slight increase in the nanotube diameter is observed. This change is attributed to the partial bulk expansion of the electrode caused by the adsorption of chloride ions. Notably, the distinctive hollow nanotube structure of CoNiPS@CF prevents significant structural cracking throughout the cycling process. This observation underscores the crucial role of this specific structure in preserving the long-term cycling stability of the electrode. And there are only a small number of crystals on the surface of the nanowires, which may be due to incomplete cleaning of the sample after cycling and the remaining salt in the solution.

Fig. 4 | **a** SAC and SEC performance for 160 desalination cycles at CoNiPS@CF in 1.0 V. **b** Specific capacity retention rate of 500 CV cycles at 50mV/s. **c** XRD of samples before and after 500 CV cycles. Electrode morphology of CoNiPS@CF (**d**) before and (**e**) after 500 CV cycles.

4. The desalination performance presented by the authors is excellent, which is very exciting! However, because CoNi-based electrodes are also one of the commonly used materials in the fields of batteries and electrocatalysis, and the materials usually have low overpotentials, it is necessary to further clarify whether there is electrolysis of water, electrocatalysis and other processes during the reaction! This is a very important factor affecting the adsorption performance of electrode materials with pseudocapacitance and EDLs as the main mechanisms.

Answer: We thank the reviewer for this comment. In this CDI system, although the maximum applied voltage value is 1.6V, due to the resistance in the battery, the actual potential difference does not meet the decomposition voltage of water (1.23 V vs. RHE), so the situation under 1.6V conditions is mainly discussed. The trace of pH was conducted to investigate possible Faradaic reactions at the CoNiPS@CF electrode surface during 75 adsorption/desorption cycles at a cell voltage of 1.6 V (Fig. 5). The results showed that pH decreased and increased as the charging and discharging steps, which is due to the lower diffusivity of bulkier Cl^- compared to that of lighter OH^- , leading to the dissociation of water and adsorption/desorption of OH^- on the electrode (in larger amounts relative to that of H^+) during charging/discharging steps, respectively (Environ. Sci. Technol., 56 (2022), 12602-12612). During an adsorption/desorption cycle, the magnitude of pH fluctuation was ca. 0.5 in the neutral condition, implying no obvious signature of chlorine and hydrogen evolution reactions in that the pH would fluctuate significantly and become alkaline if chlorine and hydrogen evolution reactions occurred. It should be noted that there was a pH decay of ca. 0.6 after 75 adsorption/desorption cycles, which is ascribable to the oxidation reactions occurred at the electrode surface in different degrees. The oxygen reduction reaction (ORR) consumed the protons, thus leading to the increase of pH in cathode region. The carbon oxidation in the anode (by reaction with water) would release the protons, decreasing pH values. The pH declines after long-term cycles indicated that the oxidation reaction in the anode is more significant than ORR in the cathode. Although different degrees of oxidation occurred in the anode and the

cathode, there was no obvious decline for desalination performance. The high retention of electrochemical capacities for the cathode and the anode contributed to the excellent cycling performance due to the partial oxidative corrosion to a low extent during 75 adsorption/desorption cycles.

Although different degrees of oxidation occurred in the anode and the cathode, there was no obvious decline for desalination performance. Thus, the electrochemical properties during long-term operation were further investigated (Fig. 4b). The retentions of specific capacities for the anode were ca. 93.4 % after 500 adsorption/desorption cycles measured by cyclic voltammograms at a scan rate of 50 mV s^{-1} . The high retention of electrochemical capacities for the anode contributed to the excellent cycling performance due to the partial oxidative corrosion to a low extent.

Fig. 5 | The pH changes versus time during 75 desalination cycles.

5. Regarding the P and S doping process, as the author mentioned, there are significant differences in the physical properties such as electronegativity and diameter of the elements, so consider discussing the potential effects of P,S-doping on surface charge. If doping alters the surface charge to a negative state, it could have a significant impact on desalination performance, possibly leading to inverted CDI. In addition, due to the new doping of different non-metal atoms, the covalent character of the metal/non-metal bond will be affected. Does the DFT can predict the property changes for Cl ion removal? How? Please explore this aspect further in your manuscript.

Answer: We thank the reviewer for this comment. To try to interpret the more intrinsic reasons for the enhancement of performance by S-doping, the density functional theory (DFT) was therefore used to study the Cl^- adsorbed energy and difference charge density of the adsorption sites in the CoNiP and CoNiPS electrodes. It is reported that the adsorption capability of Cl^- ions and the charge transfer ability to Cl^- ions at surface adsorption sites of the electrode materials are closely linked to the electrochemical reaction activity. Fig. 6 displays the optimized structural models, adsorption energy of Cl^- ions (E_{ads}), difference charge density, and Bader charge analysis of both CoNiP@CF and CoNiPS@CF. Observing the figures, CoNiPS@CF exhibits notably more negative average E_{ads} values of Cl^- ions in both Ni (-3.91 eV) and Co sites (-4.58 eV) compared to those of CoNiP@CF (-1.73 eV for Ni and -1.04 eV for Co). Additionally, CoNiPS@CF demonstrates a higher charge density at the optimal adsorption site (0.52 e for CoNiPS@CF and 0.48 e for CoNiP@CF). These findings indicate the superior capability of Cl^- ion adsorption and the charge transfer ability to Cl^- ions at the surface Ni/Co atoms of CoNiPS@CF materials. This enhancement is expected to facilitate electrochemical reactions and subsequently enhance reaction activity. Hence, alongside the kinetic advantages provided by the open structure and short diffusion paths resulting from the Kirkendall effect of the CoNiPS@CF electrode, the S-doping process significantly improves the adsorption activity of Cl^- ions.

Fig. 6 | The top view of the (a) CoNiP and (b) CoNiPS optimized structural models; c E_{ads} of Cl^- with different adsorption sites; The difference charge density and Bader charge analysis for selected (d) Ni site of CoNiP and (e) Co site of CoNiPS to Cl^- .

6. It is essential to enhance the resolution of the figures throughout the manuscript. Improved figure quality will facilitate a better understanding of the results, which are currently challenging to interpret.

Answer: We thank the reviewer for this comment. We have adjusted the resolution of the image to further improve the quality of the image to present more intuitive results.

7. In the BET characterization, the authors mention that "the pore size distribution results of

BJH show that CoNiPS@CF has a large number of microporous structures and more mesoporous structures around 10 nm than the other two electrodes, which suggests the presence of a hollow structure". However, there is no clear correlation between the presence of mesoporous structures and the hollow structure of the material. Therefore, please provide further explanation for the existence of a hollow structure.

Answer: We thank the reviewer for this comment. In fact, we cannot tell whether the structure is a hollow nanotube through the BET results, but we have provided other stronger evidence to support this conclusion. From the SEM images, we can see the obvious changes in nanoneedle diameter before and after different doping processes (Fig. 2b and 2c). The average diameter of CoNiPS (106.64 ± 16.6 nm) is much larger than the CoNiP (86.73 ± 12.94 nm). The main reason for the diameter change is the formation of hollow nanotube structures through the Kirkendall effect during the vulcanization process. The structural details of CoNiP@CF and CoNiPS@CF were investigated using TEM. Fig. 1a showed TEM images of CoNiP@CF. The EDS line scan of Fig. 1c showed that the central region element content is high, while the edges were low, indicating that the phosphorylation process basically did not change the nanowire morphology. Fig. 1b showed TEM images of CoNiPS@CF. The CoNiPS material on the surface of pCF becomes hollow, the elements in the middle and at the edges of the nanotubes was basically the same (Fig. 1d). Besides, there were nanopores on the surface of the CoNiPS nanotubes (red circles in Fig. 1c). The mechanism of the formation of the nanotube structure can be explained by the Kirkendall effect. S^{2-} first reacted with CoNiP to form a thin layer of CoNiPS, which acted as a barrier to slow down the reaction between the external S^{2-} and the internal CoNiP. Based on the non-equilibrium diffusion process between the outward CoNiP and the inward S^{2-} , voids were created in the center of the nanoneedles. As the reaction proceeds, the CoNiPS shell thickens and the CoNiP nuclei gradually decrease, eventually forming CoNiPS nanotubes; while the CoNiP nuclei present at the surface likewise become nanocenters, forming new voids on the surface of the nanotubes with external S^{2-} non-equilibrium diffusion, thus creating surface pores.

The claims "the pore size distribution results of BJH show that CoNiPS@CF has a large number of microporous structures and more mesoporous structures around 10 nm than the other two electrodes, which suggests the presence of a hollow structure" is unreasonable and has been revised in the manuscript based on the analysis above.

Fig. 7 | **a, b** TEM images and **c, d** compositional line profiles by EDS line scanning of CoNiP@CF and CoNiPS@CF.

8. Please further explain the dechlorination mechanism of the material.

Answer: We thank the reviewer for this comment. There are four main types of mechanisms for the adsorption of Cl^- by anode materials in CDI, namely electric double layer physical adsorption, conversion reaction, ion intercalation and redox reaction. The ion intercalation mechanism comes from two-dimensional materials with special structures, which can store ions between adjacent layers. LDH can accommodate anions between layers due to the positive electricity of the layers. The CV curve shows an intercalation pseudocapacitance type. In accordance with distinct reaction mechanisms, ion intercalation can be categorized into two main types: capacitor-like and battery-like mechanisms. In the capacitor-like mechanism, the CV curve appears quasirectangular and lacks discernible redox peaks. Conversely, the battery-like mechanism exhibits clear redox peaks in the CV curve, presenting an opposite profile. The CV of CoNiPS@CF presents a

quasirectangular shape without observable redox peaks indicates that CoNiPS@CF is a pseudocapacitive intercalation material like other transition metal oxides and hydroxides corroborated by GCD profiles without plateau. Besides, the nanotube array electrode is prone to be pseudocapacitive as a result of free ion diffusion (Chem. Rev. 2020, 120, 14, 6738–6782). This aligns with the open (hollow) architecture and short diffusion paths within CoNiPS@CF. A kinetic analysis can further investigate the charge storage and ion removal kinetics. It is possible to quantify the percentage of surface-controlled capacitance (capacitor-like contribution) corresponding with either a perfect diffusion-limited system (b -value of 0.5) or a perfect capacitor (b -value of 1.0). The closer it is to one, the more perfect is the pseudocapacitive response. A beneficial calculation for this consideration often referred to as Dunn analysis. For our electrode material, k_l represents 98% of the total capacity at the scan rate of $50 \text{ mV}\cdot\text{s}^{-1}$ (Fig. 8a). As a comparison, this percentage for a recently reported black phosphorus composite is 70.7% at $50 \text{ mV}\cdot\text{s}^{-1}$ (Fig. 8b). Besides, we can also derive key information via analysis of the dependency of the measured current with the sweep rate. The b -value can be determined from the plot's slope between $\log(i)$ and $\log(v)$ and indicates the charge storage mechanism. As b -values approach 0.5, the current followed a diffusion-controlled law, which was typically seen in battery-like systems. In contrast, a b -value closer to 1.0 indicated the ideal surface-controlled situation, typically seen in capacitor-like systems. As can be seen from Fig. 8c, the electrode b -values calculated for CoNiPS@CF were higher than CoNiP@CF at all voltages and all above 0.8, aligning with the pronounced pseudocapacitive behavior of the electrode material, which can provide superior rate capability and lower energy consumption in fast ion adsorption/desorption. The Trasatti analytical method was used to further analyze the electrochemical kinetics of CNPS@CF samples as shown in Fig. 8d. This method distinguished the surface control capability of the CoNiPS@CF electrode into 'internal' and 'external' surface control. The "internal" surface referred to areas that were difficult to access, whereas the "external" surface was mainly derived from surfaces directly exposed to ions and was not affected by the sweep. $q_{s,out}$ was calculated to be 72.77 mF cm^{-2} , which is 45.8% of q_s , a relatively very high value. At a scan rate of 1 mV s^{-1} , the specific capacitance of CoNiPS@CF electrode ($148.24 \text{ mF cm}^{-2}$) was 93.33% of q_s , indicating the outstanding electrochemical utilization rate of CoNiPS@CF. Thus, the CoNiPS@CF electrode provided rapid, capacitor-like ion removal and charge transfer, which can be beneficial in enhancing ion storage kinetics.

Fig. 8 | Normalized contribution ratios of surface-/diffusion-controlled capacities of (a) CoNiPS@CF and (b) CoNiP@CF. c b-values at different voltages. d The relationship between $1/q^*$ and $v^{1/2}$ and between q^* and $v^{1/2}$.

Reviewer #3 (Remarks to the Author):

The author of this article focuses on the design of chloride ion adsorption electrodes for CDI, which is a topic in urgent need of extensive research. An innovative breakthrough in this paper is to use carbon materials as conductive substrates and dope transition metal-based materials with P and S to construct a multi-hole hollow nanotube array electrodes with a multitude of highly active sites, high-speed diffusion channels and long-term cycle stability. The excellent properties of the materials introduced in this article are so impressive and this electrode proposes excellent design solutions to improve the many bottlenecks of Cl ion storage performance and mechanical properties, so it may have significant application prospects in the future. The topic is generally interesting to the potential readers in this journal but there are some issues need to be addressed to fully support the claims in this article. Thus, a required minor revision is recommended before consideration for publication.

1. This article used a static CDI electrode, and ion exchange membranes separated the saline chamber and the electrodes. For carbon felt base materials, they are often used in flow-through configurations. I don't understand how to calculate the desalination capacity of active materials loaded on carbon felt fibers in flow-by mode? How to calculate the material loading of transition metal-based materials in carbon felt? In data presentation, would it be more appropriate to use volumetric desalting capacity?

Answer: We thank the reviewer for this comment. In our flow-by hybrid CDI system, the anode is the CoNiPS@CF, and the cathode is the activated carbon electrode. The configuration diagram is shown in Fig. 1. We obtain the load mass by subtracting the mass of the pretreated carbon felt (pCF) from the mass of the final sample. Specifically, weigh the dried pCF to obtain the mass of pCF. And after the hydrothermal process is completed, the sample is simply washed and dried in drying oven, and the dried sample is weighed to obtain the total mass of the final sample (like CoNiPS@CF). Therefore, the load mass of CoNiPS can be obtained by subtracting the two masses and the loading capacity is obtained by dividing the load mass of CoNiPS by the area of the single pCF ($3\text{ cm} \times 3\text{ cm} \times 0.1\text{ cm}$, the area is $3 \times 3 \times 2 + 3 \times 0.1 \times 4 = 19.2\text{ cm}^2$).

The removal capacity of NaCl based on the prepared electrode area were calculated as shown below:

$$\text{desalination capacity}(Q) = \frac{(C_0 - C_e)V}{M} \text{ (mg of NaCl/g of CoNiPS@CF)}$$

where C_0 and C_e are the NaCl concentration (mg L^{-1}) at initial and equilibrium stages, respectively, V is the total NaCl volume(L), M is the loading mass of CoNiPS in one piece of CF.

In this system, the concept of volumetric desalting capacity is not applicable. Despite the three-dimensional nature of the carbon felt electrode, both the activation process and active material loading occur only on the surface of the carbon felt. The adsorption capacity of both CF and pCF is low. Therefore, it can be considered that the desalination process primarily takes place on the surface of the electrode. Furthermore, as it operates in a flow-by system, the saltwater cannot permeate the entire three-dimensional space of the electrode. The effective contact area is limited to the interface between the carbon felt and the anion exchange membrane. Hence, the consideration of volumetric desalting capacity is not applicable in this context.

Fig. 1 | Configuration diagram of CDI system.

2. After P doping, according to the author's description, it will not have a significant impact on the microstructure of the electrode surface. According to Figure S2, the CoNiP@CF sample has more microporous structure. The reason for this doesn't seem to be discussed in the manuscript. In addition, the presence of the CoNiS@CF sample in the desalination test is very reasonable, but why was the N₂ adsorption/desorption isotherms and specific surface area not tested for this sample? This can further illustrate the influence of S doping on its micromorphology.

Answer: We thank the reviewer for this comment. In this experiment, based on the CoNiOH material synthesized by hydrothermal reaction, phosphorus doping, and sulfur doping were

gradually performed through chemical vapor deposition (CVD) method, and finally CoNiP and CoNiPS were obtained. The advantage of the CVD method is that it has little change in the original electrode morphology and can maintain its original high specific surface area nanoneedle morphology. Through SEM morphology and diameter analysis (Fig. 2a, 2b and 2c), we can also see that compared to CoNiOH, the nanoneedle morphology of CoNiP has basically remained unchanged. And the nanoneedle diameters of the two are also very close (CoNiOH: 85.28 ± 14.11 nm, CoNiP: 86.73 ± 12.94 nm). However, when we further doped sulfur, the surface nanoneedle morphology changed and CoNiPS has a significantly thicker diameter, which means porous nanotubes were gradually formed from the nanoneedles. This is due to the significant Kirkendall phenomenon in the sulfur doping process. S^{2-} first reacted with CoNiP to form a thin layer of CoNiPS, which acted as a barrier to slow down the reaction between the external S^{2-} and the internal CoNiP. Based on the non-equilibrium diffusion process between the outward CoNiP and the inward S^{2-} , voids were created in the center of the nanowires. As the reaction proceeds, the CoNiPS shell thickens and the CoNiP nuclei gradually decrease, eventually forming CoNiPS nanotubes. In addition, studies have shown that P doping can also produce the Kirkendall effect (Materials Today Nano, 2022, 18: 100195.) and have a higher specific surface area. Based on the results of this experiment (Fig. 2d and 2e), it is evident that P doping does indeed alter the pore size distribution to some extent. But it cannot produce a strong enough Kirkendall effect to create a cavity structure. Its primary role lies in imparting metalloidal characteristics and excellent electrical conductivity to the electrode. In addition, we also added BET analysis of CoNiS samples. The results show that the CoNiS sample has a larger specific surface area ($59.08 \text{ m}^2 \text{ g}^{-1}$) than CoNiP ($50.135 \text{ m}^2 \text{ g}^{-1}$), which also confirms that its Kirkendall effect is stronger, that is, the diffusion rate difference between S and the metal source is larger than that of P.

Fig. 2 | Diameter distribution of (a) CoNiOH@CF, (b) CoNiP@CF nanoneedles and (c) CoNiPS@CF nanotubes from SEM. d N₂ adsorption and desorption curves. e BJH pore size analysis of different electrodes.

3. The author has to show the cycled SAC-time profiles and the corresponding current-time curves for various simples. For most transition metal materials, there may be desorption difficulties in the electrode, and this irreversible adsorption has a negative impact on the cycle performance of the electrode. is extremely disadvantageous. The authors need to provide more data to prove this.

Answer: We thank the reviewer for this comment. We have incorporated data illustrating the evolution of conductivity and current over time in a NaCl solution with an initial concentration of ~1000 mg/L under a constant voltage of 1.2V for three cycles (Fig. 3). Compared to the other two samples, the desorption capacity of CoNiP is notably smaller than its adsorption capacity. This is due to the higher adsorption energy of Cl⁻ by CoNiP, resulting in irreversible capacity loss and a high ion diffusion barrier. Consequently, active adsorption sites may gradually deactivate during the experiment, diminishing the electrode's adsorption capacity. This results in the inability of the electrode to sustain optimal performance over an extended period, thereby reducing cycle stability. Furthermore, ion adsorption will induce a certain degree of bulk phase expansion. The fact that the desorption capacity is lower than the adsorption capacity indicates that certain ions on the surface are adsorbed into the electrode layers and cannot be effectively

desorbed. This exacerbates bulk phase expansion, which in turn impacts the structural stability of the electrode. In contrast, CoNiPS@CF electrode showed a decent reversibility, and the desorption capacity and adsorption capacity are basically equal, which can be attributed to the lower Cl^- adsorption energy of CoNiPS@CF. And it has the highest specific adsorption capacity, with an average SAC of $71.4 \pm 4.3 \text{ mg}_{\text{Cl}^-} \text{ g}^{-1}$.

Fig. 3 | Conductivity and current versus time images of (a) CoNiOH@CF, (b) CoNiP@CF and (c) CoNiPS@CF

4. Water splitting is one of the common side reactions in the CDI process, especially at high voltage. The test condition of the highest desalination capacity in this article reaches 1.8 V, which is a very high application value in transition metal materials because of their generally low overpotential. And at 1.8 V, the energy consumption of the electrode increases significantly, which may be one of the signs of water electrolysis. The authors need further discussion to prove that desalination can be performed at 1.8 V.

Answer: We thank the reviewer for this comment. In this CDI system, the highest applied voltage value is 1.6V, so the situation under 1.6V conditions is mainly discussed. The trace of pH was conducted to investigate possible Faradaic reactions at the CoNiPS@CF electrode surface during 75 adsorption/desorption cycles at a cell voltage of 1.6 V (Fig. 4a). The results showed that pH decreased and increased as the charging and discharging steps, which is due to the lower diffusivity of bulkier Cl^- compared to that of lighter OH^- , leading to the dissociation of water and adsorption/desorption of OH^- on the electrode (in larger amounts relative to that of H^+) during charging/discharging steps, respectively (Environ. Sci. Technol., 56 (2022), 12602-12612). During an adsorption/desorption cycle, the magnitude of pH fluctuation was ca. 0.6 in the neutral condition, implying no obvious signature of chlorine and hydrogen evolution reactions in that the pH would fluctuate significantly and become alkaline if chlorine and hydrogen evolution reactions occurred. It should be noted that there was a pH decay of ca. 0.6 after 75 adsorption/desorption cycles, which is ascribable to the oxidation reactions occurred at the electrode surface in different degrees. The oxygen reduction reaction (ORR) consumed the protons, thus leading to the increase of pH in cathode region. The carbon oxidation in the anode (by reaction with water) would release the protons, decreasing pH values. The pH declines after long-term cycles indicated that the oxidation reaction in the anode is more significant than ORR in the cathode. Although different degrees of oxidation occurred in the anode and the cathode, there was no obvious decline for desalination performance. The high retention of electrochemical capacities for the cathode and the anode contributed to the excellent cycling performance due to the partial oxidative corrosion to a low extent during 75 adsorption/desorption cycles.

Although different degrees of oxidation occurred in the anode and the cathode, there was no obvious decline for desalination performance. Thus, the electrochemical properties during long-term operation were further investigated (Fig. 4b). The retentions of specific capacities for the anode were ca. 93.4 % after 500 adsorption/desorption cycles measured by cyclic voltammograms at a scan rate of 50 mV s^{-1} . The high retention of electrochemical capacities for the anode contributed to the excellent cycling performance due to the partial oxidative corrosion to a low extent.

Fig. 4 | **a** The pH change versus time during 75 desalination cycles. **b** Capacitance retention rate of cyclic voltammograms at a scan rate of 50 mV s^{-1}

5. In electrochemical tests, why is the capacitance of the sulfur-doped sample lower than that of the nearly phosphorus-doped sample? This does not seem to fully correspond to the SAC data. Furthermore, why is the shape of the curve different between the long cycle CV data and the individual tests? Is there a difference in the current response between the two? The author needs further explanation.

Answer: We thank the reviewer for this comment. The calculated values of capacitance at different sweep rates were shown in Fig. 5a. In the process of increasing the sweep speed, CoNiPS@CF can retain 42.95% of its original capacity, while CoNiP@CF retains only 25.50%. This may be because the CoNiP@CF sample is unstable and easily oxidized to produce a passivation layer, which can also be derived from the XPS analysis results: the P to PO peak ratio changed from 6.8% to 13.3% after sulfur doping (Fig. 5b), suggesting that the presence of sulfur reduced the formation of the CoNiP@CF passivation layer. Besides, CoNiOH could generate from the reaction of CoNiP nanoneedles and electrolyte and cause a rapid decrease in the specific capacitance value (Adv.

Funct. Mater., 25: 7530-7538.). Therefore, the CoNiPS electrode has better stability and conductivity, and its specific capacitance value decreases more slowly as the scanning rate increases. During the CV test, it is evident that the specific capacitance performance of the CoNiPS@CF sample surpasses that of other samples at high scanning rates. This outcome suggests that the CoNiPS@CF sample possesses a greater number of surface-controlled active sites, enabling it to maintain heightened specific capacitance performance during high-speed scanning rates. This was related to the unique hollow nanotube structure of CoNiPS@CF, indicating that the electrode had a highly open pore structure with extremely accessible active sites, which facilitated the high rate of ion adsorption and desorption on the electrode surface (Angew Chem Int Ed Engl, 58 (2019), 13840-13844). Consequently, the electrode attains a superior desalination capacity at high current, indicating an elevated specific adsorption rate. This trend is also observable in the proportion of surface and diffusion capacitance at different scanning rates. Across various scan rates, the CoNiPS@CF samples consistently exhibit a higher proportion of surface-controlled capacitance compared to CoNiP@CF (Fig. 5d and e). Moreover, the calculated b-values for CoNiPS@CF consistently surpass those for CoNiP@CF at all voltages, consistently exceeding 0.8 (Fig. 5f). This suggests that CoNiPS@CF demonstrates more pronounced pseudocapacitive properties at all voltages, offering superior rate capability and lower energy consumption during rapid ion adsorption/desorption processes (Nat Mater, 16 (2017), 454-460).

CV tests carried out for the CoNiPS@CF electrode at various sweep rates all showed a leaf shape with no prominent redox peaks (Fig. 5c), demonstrating the excellent capacitive reversibility of the CoNiPS@CF electrode; and the CV curves of the CoNiP@CF electrode were similar to those of CoNiPS@CF (25 mV s⁻¹), indicating that the storage mechanism of Cl⁻ for both electrodes can be attributed to pseudocapacitance. For the long cycle test of CV, it was carried out at 10 mV/s. For CV curves at different sweep speeds, their shapes will be slightly different, and both CV images are leaf-shaped, which both shows that it has a pseudocapacitive adsorption mechanism.

Fig. 5 | **a** specific capacity comparison of CoNiOH@CF, CoNiP@CF and CoNiPS@CF. **b** XPS high-resolution spectra of P 2p for CoNiPS@CF and CoNiP@CF. **c** CV curves of CoNiPS@CF. Normalized contribution ratios of surface-/diffusion-controlled capacities of **(d)** CoNiP@CF and **(e)** CoNiPS@CF. **f** b-values at different voltages of CoNiP@CF and CoNiPS@CF.

6. In Figure 4g, there is a certain gap between the NaCl removal amount and the previous SAC value, but it seems that no significant difference in reaction conditions can be seen. Please explain the difference in SAC values, or add relevant experimental conditions and SAC-time data.

Answer: We thank the reviewer for this comment. Activated carbon (AC) and the prepared material were used as the cathode and anode of the CDI system with 1000 mg L⁻¹ NaCl, respectively. **Fig. 4a** illustrated the variation of the concentration of each solution in a single cycle under constant voltage (1.2 V) conditions. As shown in Fig. 4b, the specific adsorption capacity (SAC) of the CoNiPS@CF electrode is as high as 71.4 ± 4.3 mg_{Cl⁻} g⁻¹, which was the highest desalination capacity of the individual electrodes and corresponded to the superior capacitive performance of CoNiPS@CF. Correspondingly, CoNiPS@CF exhibited the highest specific adsorption rate, averaging 2.38 ± 0.14 mg_{Cl⁻} g⁻¹ min⁻¹ over 30 minutes. This was attributed to the short Cl⁻ transport path and the higher ion adsorption rate facilitated by the highly open multi-

hollow nanotube structure of CoNiPS@CF, providing a large number of surface-accessible active sites compared to other electrodes. This characteristic suggests promising applications for the electrode in brackish water desalination. To evaluate its feasibility, we conducted treatment experiments using a single 3×3 cm sized CoNiPS@CF electrode to treat 20 ml of brackish water at 1.2 V. The results indicate that after the fourth cycle, lasting approximately 100 minutes, the Cl^- concentration in the brackish water meets drinking water standards, demonstrating successful seawater desalination. To compare the desalination performance of the electrodes in the two experiments, we calculated the desalination capacity at each cycle in the brackish water experiment and determined the average Specific Adsorption Capacity (SAC). The average SAC of brackish water experiment is $67.83 \text{ mg}_{\text{Cl}^-} \text{ g}^{-1}$, which is similar with CDI system ($71.4 \text{ mg}_{\text{Cl}^-} \text{ g}^{-1}$). The difference in the actual decrease in brackish water concentration in a single cycle is mainly due to the following reasons: 1) The actual electrode loading capacity is different. Since the active material is grown in situ on the surface of the carbon felt electrode by the hydrothermal process, and it is difficult to control the loading amount to be exactly the same in the hydrothermal process, this process will result in different electrode loading amounts (Nano Energy, 39(2017), 162-171). In the CDI test, the CoNiPS@CF electrode loading capacity was 26.95 mg, while the loading capacity in the brackish water experiment was 16.48 mg, so the actual saline concentration drop in the brackish water experiment is slightly smaller; 2) in the CDI system, the concentration versus time curve is for the third cycle, while the brackish water experiment is recorded from the first cycle. An activation process occurs during the use of the electrode, potentially leading to lower capacity in the initial cycles. This phenomenon may contribute to a lower SAC value in the brackish water experiment during the initial cycles.

Fig. 6 | **(a)** The change of NaCl concentration versus time during CDI process and **(b)** SAC and SAR of different electrodes. **(c)** Plot of salt concentration change under multiple cycles in brackish water experiment and **(d)** average SAC of the four cycles.

7. Why didn't the author compare the properties of CoNiOH samples in electrochemical tests? It remains somewhat unclear about the improvement in electrochemical performance described by the authors after doping.

Answer: We thank the reviewer for this comment. In order to have a better understanding of improvement in electrochemical performance after doping, we tested the CV, GCD and EIS electrochemical properties of CoNiOH sample compared them with the CoNiP and CoNiPS samples. Electrochemical tests were carried out using a three-electrode system in a 1 mol L⁻¹ NaCl solution. CV tests were carried out for the three electrodes at various sweep rates, and the findings (**Fig. 7a**) indicated that following doping, both CoNiPS@CF and CoNiP@CF exhibit superior electrochemical performance compared to CoNiOH. This improvement may be attributed to the modification of conductivity and active adsorption sites in the material through P and S doping, leading to a substantial increase in the specific capacitance of the electrode. At low sweep rates, the CoNiP@CF electrode had a higher capacitance compared to the CoNiPS@CF electrode (e.g. 209.86 mF cm⁻² at 1 mV s⁻¹); however, in the process of increasing the sweep speed, CoNiPS@CF can retain 42.95% of its original capacity, while CoNiP@CF retains only 25.50%. This was related to the unique hollow nanotube structure of CoNiPS@CF, indicating that the electrode had a highly open pore structure with extremely accessible active sites, which facilitated the high rate of ion adsorption and desorption on the electrode surface. Similar conclusions can be drawn from Galvanostatic Charge-Discharge (GCD) tests performed at a current density of 1 mA cm⁻² (**Fig. 7b**). CoNiPS@CF exhibits the highest specific capacity, measuring at 0.103 mAh cm⁻². Furthermore, no plateau was observed in any of the samples, suggesting that all the samples work through a pseudocapacitive mechanism. CoNiPS@CF also demonstrated higher conductivity compared to CoNiP@CF and CoNiOH@CF, as shown by the Nyquist plot in **Fig. 7c**. In detail, compared to CoNiP@CF (3.704 Ω) and CoNiOH@CF(4.722 Ω), CoNiPS@CF showed a lower internal resistance (R_{int}) value (3.082 Ω), indicating a lower percentage of charge consumption and higher charging efficiency. The charge transfer resistance (R_{ct}) value of the CoNiPS@CF electrode (3.386 Ω) was also lower than that of CoNiP@CF (3.882 Ω) and CoNiOH@CF(7.666 Ω), implying that CoNiPS@CF had better conductivity and superior electrochemical kinetics. Therefore, P doping does contribute to a significant improvement in electrode conductivity.

Moreover, the conductivity of the electrode continues to improve to a certain extent after sulfur doping, possibly due to the reduction in the content of the P-O passivation layer after sulfur doping, consequently enhancing the overall conductivity of the electrode.

Fig 7 | **a** The comparison of specific capacity with different samples. **b** The galvanostatic charge-discharge (GCD) profiles of different samples at 1 mA cm^{-2} . **c** Nyquist plots of different samples.

REVIEWERS' COMMENTS

Reviewer #1 (Remarks to the Author):

The authors have addressed comments raised in first review report. The revision of the manuscript significantly enhanced the quality of manuscript and therefore I recommend its publication without further revision.

Reviewer #2 (Remarks to the Author):

The issues raised last time have been addressed nicely and this manuscript was agreed to be accepted.

Reviewer #3 (Remarks to the Author):

After careful evaluation of the revised manuscript, I recommend this manuscript should be accepted by Nature Communications.